# A Spacetime Meshless Method for Modeling Subsurface Flow with a Transient Moving Boundary

**Cheng-Yu Ku** [1,2]📧, **Chih-Yu Liu** [1,*]📧, **Jing-En Xiao** [1]📧, **Weichung Yeih** [1,2]📧 and **Chia-Ming Fan** [1,2]📧

1  Department of Harbor and River Engineering, National Taiwan Ocean University, Keelung 20224, Taiwan; chkst26@mail.ntou.edu.tw (C.-Y.K.); 20452002@email.ntou.edu.tw (J.-E.X.); wcyeih@mail.ntou.edu.tw (W.Y.); cmfan@ntou.edu.tw (C.-M.F.)
2  Center of Excellence for Ocean Engineering, National Taiwan Ocean University, Keelung 20224, Taiwan
*  Correspondence: 20452003@email.ntou.edu.tw; Tel.: +886-2-2462-2192 (ext. 6159)

**Abstract:** In this paper, a spacetime meshless method utilizing Trefftz functions for modeling subsurface flow problems with a transient moving boundary is proposed. The subsurface flow problem with a transient moving boundary is governed by the two-dimensional diffusion equation, where the position of the moving boundary is previously unknown. We solve the subsurface flow problems based on the Trefftz method, in which the Trefftz basis functions are obtained from the general solutions using the separation of variables. The solutions of the governing equation are then approximated numerically by the superposition theorem using the basis functions, which match the data at the spacetime boundary collocation points. Because the proposed basis functions fully satisfy the diffusion equation, arbitrary nodes are collocated only on the spacetime boundaries for the discretization of the domain. The iterative scheme has to be used for solving the moving boundaries because the transient moving boundary problems exhibit nonlinear characteristics. Numerical examples, including harmonic and non-harmonic boundary conditions, are carried out to validate the method. Results illustrate that our method may acquire field solutions with high accuracy. It is also found that the method is advantageous for solving inverse problems as well. Finally, comparing with those obtained from the method of fundamental solutions, we may obtain the accurate location of the nonlinear moving boundary for transient problems using the spacetime meshless method with the iterative scheme.

**Keywords:** spacetime meshless method; Trefftz functions; transient; moving boundary; nonlinear

## 1. Introduction

The free surface flow in soils can be defined as moving boundary problems because the location of one or more of the domain boundaries is unknown [1–4]. The phreatic line is located between the fluid phase and the air phase of the soil. It is sometimes regarded as the phase change problem [5–7]. Phase change problems are often encountered in engineering, industry, and problems such as the design of roadways in cold regions [8–10]. These problems are usually non-stationary as well as nonlinear due to the phase change depending on time and complicated material properties [11]. Therefore, great challenges may be raised for solving the problems using analytical solutions.

Various numerical approaches [12], such as the boundary element method [13], the interpolation finite difference method [14], the finite element method [15], the finite volume method [16], the local radial basis function collocation method [17], the method of approximate particular solutions [18], and the method of fundamental solutions (MFS) [19,20], have been utilized for dealing with moving boundary problems. The collocation method can be categorized into one of the meshless methods [21,22].

The discretization of the domain for the collocation approaches is relatively simple because the arbitrary points are assigned only on the boundary if we find the basis functions, which must satisfy the governing equation [23,24]. This idea was developed by Erich Trefftz and known as the Trefftz method [25]. The Trefftz method is widely adopted for dealing with the boundary value problem (BVP) [26–29]. This method is often found to solve the Laplace-type equations. The reason is that the derivation of the Trefftz functions for other partial differential equations may be very challenging [30]. Previous studies have demonstrated that applications of the Trefftz method may be limited to linear as well as stationary problems. Recently, a study on solving subsurface flow problems with free and moving boundaries governed by the Laplace governing equation adopting the Trefftz method has been developed [23]. However, the engineering applications of the Trefftz method with complete Trefftz functions for dealing with transient problems are still hardly found, where solving transient moving boundary problems adopting the Trefftz method rarely exist.

In this study, we propose the spacetime meshless approach using Trefftz functions for solving subsurface flow problems with a transient moving boundary. The subsurface flow problem with a transient moving boundary is governed by the two-dimensional diffusion equation, where the position of the nonlinear moving boundary is previously unknown. We solve the subsurface flow problems utilizing the Trefftz method, in which the Trefftz basis functions can be obtained from the general solutions using the separation of variables. We propose the spacetime collocation scheme such that the solutions of the governing equation are approximated numerically by the superposition theorem using the proposed basis functions, which match the data at spacetime boundary collocation points. Because the proposed basis functions fully satisfy the diffusion equation, arbitrary nodes are collocated only on the spacetime boundaries for the discretization of the domain. Because the transient moving boundary problems exhibit nonlinear characteristics, the iterative scheme has to be used for solving the moving boundaries. Numerical examples, including harmonic and non-harmonic boundary conditions, were carried out to validate the method. The derivation of the spacetime collocation scheme utilizing Trefftz functions is depicted in the following section.

## 2. The Governing Equation

The transient moving boundary phenomenon is governed by the two-dimensional diffusion governing equation in the dimensionless form in polar coordinates as follows,

$$\frac{\partial^2 h(r,\theta,t)}{\partial r^2} + \frac{1}{r}\frac{\partial h(r,\theta,t)}{\partial r} + \frac{1}{r^2}\frac{\partial^2 h(r,\theta,t)}{\partial \theta^2} = \frac{\partial h(r,\theta,t)}{\partial t} \text{ in } \mathfrak{R}_t, \tag{1}$$

where $\mathfrak{R}_t$ is the spacetime domain of the transient moving boundary problems, $h$ is the total head, $\theta$ denotes the polar angle, $r$ is the dimensionless variable, which is expressed as $r = \hat{r}/R_0$, $\hat{r}$ is the radius, $R_0$ is the maximum dimension of the problem, which is also named as the length of characteristic, $t$ is the dimensionless variable, which is expressed as $t = \hat{t}T/R_0^2 S$, $S$ is the storage coefficient, $\hat{t}$ is the time, and $T$ is the transmissibility coefficient. While $\hat{r}$ is in the range of $0 < \hat{r} < R_0$, $r$ is in the range of $0 < r < 1$. The initial condition for solving Equation (1) is as follows,

$$h(r,\theta,t=0) = h_0(r,\theta,t=0), \tag{2}$$

where $h_0$ is the initial total head. To solve Equation (1), the boundary data are expressed as

$$h(r,\theta,t) = D(r,\theta,t) \text{ on } \Gamma_D^t, \tag{3}$$

$$h_n(r,\theta,t) = \frac{\partial h(r,\theta,t)}{\partial n} = N(r,\theta,t) \text{ on } \Gamma_N^t, \tag{4}$$

where $\Gamma_D^t$ is the spacetime Dirichlet boundary condition, $\Gamma_N^t$ is the spacetime Neumann boundary condition, the subscript $D$ is the Dirichlet boundary data, the subscript $N$ denotes the Neumann

boundary data, the subscript $n$ is the outward normal direction, $D(r, \theta, t)$ and $N(r, \theta, t)$ are the Dirichlet and Neumann boundary data of the transient moving boundary problems in the spacetime domain, respectively.

## 3. The Spacetime Meshless Method Using Trefftz Functions

### 3.1. Trefftz Functions for Transient Moving Boundary Problems

The spacetime meshless method using Trefftz functions is rooted in the Trefftz method. Thus, it is necessary for the nonlinear moving boundary problems to formulate the general solutions. To formulate the transient Trefftz functions for the nonlinear moving boundary problems, the separation of variables is adopted.

$$h(r, \theta, t) = \varphi(r, \theta)\Omega(t), \tag{5}$$

where $\varphi(r, \theta)$ and $\Omega(t)$ are functions. The total head $h(r, \theta, t)$ is a product of two functions. For simplicity, the following notations are considered.

$$\varphi_r = \frac{d\varphi(r, \theta)}{dr}, \; \varphi_{rr} = \frac{d^2\varphi(r, \theta)}{dr^2}, \; \varphi_{\theta\theta} = \frac{d^2\varphi(r, \theta)}{d\theta^2} \text{ and } \Omega' = \frac{d\Omega(t)}{dt}, \tag{6}$$

where the subscript $r$ denotes the first derivative with respect to $r$, the subscript $rr$ denotes the second derivative with respect to $r$, the subscript $\theta\theta$ denotes the second derivative with respect to $\theta$. Inserting Equation (5) into Equation (1), by taking into account notation Equation (6), we have

$$(\varphi_{rr} + \frac{1}{r}\varphi_r + \frac{1}{r^2}\varphi_{\theta\theta})\Omega = \varphi\Omega'. \tag{7}$$

We further consider the following equation

$$\varphi(r, \theta) = R(r)W(\theta), \tag{8}$$

where $R(r)$ and $W(\theta)$ are functions. The function $\varphi(r, \theta)$ is a product of two functions, including $R(r)$ and $W(\theta)$. Each function depends only on one of the variables $r$ or $\theta$. Inserting Equation (8) into Equation (7), we obtain

$$R''W\Omega + \frac{1}{r}R'W\Omega + \frac{1}{r^2}RW''\Omega = RW\Omega', \tag{9}$$

where $R' = \frac{dR(r)}{dr}$, $R'' = \frac{d^2R(r)}{dr^2}$, $W'' = \frac{d^2W(\theta)}{d\theta^2}$, and $\Omega' = \frac{d\Omega(t)}{dt}$.

Dividing by $R(r)W(\theta)\Omega(t)$ on both sides in Equation (9), we can then obtain

$$\begin{cases} \frac{\Omega'}{\Omega} = \lambda, \\ \frac{r^2R'' + rR' - r^2R\lambda}{R} = \chi \\ -\frac{W''}{W} = \chi, \end{cases} \tag{10}$$

where $\lambda$ and $\chi$ are separation constants. We introduce $p$ and $q$ and assume that $\lambda = p^2$ or $\lambda = -p^2$ and $\chi = q^2$ or $\chi = -q^2$ to ensure $\lambda$ and $\chi$ to be positive or negative value, respectively. The formulation of Trefftz functions for transient moving boundary problems are expressed in the following description. Considering the combination of positive or negative values for $\lambda$ and $\chi$, there are six possible scenarios. If we consider the first scenario, $\lambda = 0$ and $\chi = q^2$, we may obtain

$$\begin{cases} \Omega(t) = A_1, \\ R(r) = A_2 r^q + A_3 r^{-q}, \\ W(\theta) = A_4 \cos(q\theta) + A_5 \sin(q\theta), \end{cases} \tag{11}$$

where $A_1$, $A_2$, $A_3$, $A_4$, and $A_5$ are arbitrary constants that have to be evaluated. Inserting Equation (11) into Equation (5) may yield

$$h(r,\theta,t) = \overline{A}_1 r^q \cos(q\theta) + \overline{A}_2 r^q \sin(q\theta) + \overline{A}_3 r^{-q} \cos(q\theta) + \overline{A}_4 r^{-q} \sin(q\theta), \tag{12}$$

where $\overline{A}_1$, $\overline{A}_2$, $\overline{A}_3$, and $\overline{A}_4$ are arbitrary constants that have to be evaluated. We may find solutions for five other scenarios including $\lambda = 0$ and $\chi = 0$, $\lambda = p^2$ and $\chi = q^2$, $\lambda = p^2$ and $\chi = 0$, $\lambda = -p^2$ and $\chi = q^2$, and $\lambda = -p^2$ and $\chi = 0$ using the same procedure, as listed in Appendix A. As a result, we may obtain the complete Trefftz functions described as follows,

$$\mathbf{T} = \left\{ \overline{T}_1, \overline{T}_2, \overline{T}_3, \ldots, \overline{T}_{18} \right\}, \tag{13}$$

where $\mathbf{T}$ denotes the Trefftz basis functions, and $\overline{T}_1, \overline{T}_2, \overline{T}_3, \ldots, \overline{T}_{18}$ denotes the functions, as listed in Appendix A. The transient numerical solution for the two-dimensional subsurface flow problem with a transient moving boundary is expressed by the series expansion as follows,

$$
h(r,\theta,t) = \overline{a} + \overline{b}\ln r + \sum_{q=1}^{v} \left\{ \begin{array}{l} \overline{c}_{1q} r^q \cos(q\theta) + \overline{c}_{2q} r^q \sin(q\theta) + \overline{c}_{3q} e^{p^2 t} I_0(qr) + \overline{c}_{4q} e^{-p^2 t} J_0(qr) \\ + \overline{c}_{5q} r^{-q} \cos(q\theta) + \overline{c}_{6q} r^{-q} \sin(q\theta) + \overline{c}_{7q} e^{p^2 t} K_0(qr) + \overline{c}_{8q} e^{-p^2 t} Y_0(qr) \end{array} \right\}
$$
$$
+ \sum_{p=1}^{v} \left\{ \begin{array}{l} \overline{d}_{1qp} e^{p^2 t} I_q(pr) \cos(q\theta) + \overline{d}_{2qp} e^{p^2 t} I_q(pr) \sin(q\theta) \\ + \overline{d}_{3qp} e^{-p^2 t} J_q(pr) \cos(q\theta) + \overline{d}_{4qp} e^{-p^2 t} J_q(pr) \sin(q\theta) \\ + \overline{d}_{5qp} e^{p^2 t} K_q(pr) \cos(q\theta) + \overline{d}_{6qp} e^{p^2 t} K_q(pr) \sin(q\theta) \\ + \overline{d}_{7qp} e^{-p^2 t} Y_q(pr) \cos(q\theta) + \overline{d}_{8qp} e^{-p^2 t} Y_q(pr) \sin(q\theta) \end{array} \right\}, \tag{14}
$$

where $v$ denotes the order of the Trefftz functions, and $\overline{a}$, $\overline{b}$, $\overline{c}_{1q} \ldots, \overline{d}_{8qp}$ denote unknown coefficients, $I_0$ and $I_q$ denote the modified Bessel functions of the first kind of zero order and of $q$ order, respectively. $J_0$ and $J_q$ denote the Bessel functions of the first kind of zero order and of $q$ order, respectively. $K_0$ and $K_q$ denote the modified Bessel functions of the second kind of zero order and of $q$ order, respectively. $Y_0$ and $Y_q$ denote the Bessel functions of the second kind of zero order and of $q$ order, respectively.

For the infinite domain or domain with cavities, the Trefftz basis functions are described as

$$
h(r,\theta,t) = \overline{b}\ln r + \sum_{q=1}^{v} \left\{ \overline{c}_{5q} r^{-q} \cos(q\theta) + \overline{c}_{6q} r^{-q} \sin(q\theta) + \overline{c}_{7q} e^{p^2 t} K_0(pr) + \overline{c}_{8q} e^{-p^2 t} Y_0(pr) \right.
$$
$$
\left. + \sum_{p=1}^{v} \left\{ \begin{array}{l} \overline{d}_{5qp} e^{p^2 t} K_q(pr) \cos(q\theta) + \overline{d}_{6qp} e^{p^2 t} K_q(pr) \sin(q\theta) \\ + \overline{d}_{7qp} e^{-p^2 t} Y_q(pr) \cos(q\theta) + \overline{d}_{8qp} e^{-p^2 t} Y_q(pr) \sin(q\theta) \end{array} \right\} \right\}. \tag{15}
$$

When the domain is simply connected, we may consider only positive basis functions. Consequently, the above equation is simplified as the following equation.

$$
h(r,\theta,t) = \overline{a} + \sum_{q=1}^{v} \left\{ \overline{c}_{1q} r^q \cos(q\theta) + \overline{c}_{2q} r^q \sin(q\theta) + \overline{c}_{3q} e^{p^2 t} I_0(qr) + \overline{c}_{4q} e^{-p^2 t} J_0(qr) \right.
$$
$$
\left. + \sum_{p=1}^{v} \left\{ \begin{array}{l} \overline{d}_{1qp} e^{p^2 t} I_q(pr) \cos(q\theta) + \overline{d}_{2qp} e^{p^2 t} I_q(pr) \sin(q\theta) \\ + \overline{d}_{3qp} e^{-p^2 t} J_q(pr) \cos(q\theta) + \overline{d}_{4qp} e^{-p^2 t} J_q(pr) \sin(q\theta) \end{array} \right\} \right\}. \tag{16}
$$

To evaluate the unknown coefficients of $\overline{a}$, $\overline{c}_{1q}, \ldots, \overline{d}_{4qp}$ in Equation (16), the spacetime collocation scheme must be utilized. Using the spacetime collocation scheme and applying the Dirichlet boundary data in Equation (16), a system of equations may then be yielded.

$$
\begin{bmatrix}
1 & r_1^q\cos(q\theta_1) & r_1^q\sin(q\theta_1) & \cdots & e^{-p^2t_1}J_q(pr_1)\sin(q\theta_1) \\
1 & r_2^q\cos(q\theta_2) & r_2^q\sin(q\theta_2) & \cdots & e^{-p^2t_2}J_q(pr_2)\sin(q\theta_2) \\
\vdots & \vdots & \vdots & \ddots & \vdots \\
1 & r_s^q\cos(q\theta_s) & r_s^q\sin(q\theta_s) & \cdots & e^{-p^2t_s}J_q(pr_s)\sin(q\theta_s)
\end{bmatrix}
\begin{bmatrix}
\bar{a} \\ \bar{c}_{1q} \\ \vdots \\ \bar{d}_{4qp}
\end{bmatrix}
=
\begin{bmatrix}
h_1 \\ h_2 \\ \vdots \\ h_s
\end{bmatrix},
\tag{17}
$$

where $t_1,\ t_2,\cdots,t_s$ are time in dimensionless form, $r_1,\ r_2,\cdots,r_s$ are radiuses in dimensionless form, $\theta_1,\ \theta_2,\cdots,\theta_s$ are polar angles in dimensionless form, $h_1,\ h_2,\cdots,h_s$ are Dirichlet boundary data, the subscript $s$ denotes the number of boundary points, and $\bar{a},\ \bar{c}_{1q},\cdots,\bar{d}_{4qp}$ denote the unknown coefficients. Equation (17) is expressed as

$$
\mathbf{Hy} = \mathbf{Z},
\tag{18}
$$

where $\mathbf{H}$ denotes a matrix of the Trefftz basis functions with the size of $s \times w$, $\mathbf{y}$ denotes a vector of the unknown coefficients with the size of $w \times 1$, $\mathbf{Z}$ denotes a vector of accessible boundary value at boundary collocation points with the size of $s \times 1$, $s$ denotes the number of boundary points, $w$ denotes the term related to the order of the Trefftz basis function. Solving Equation (18), we may acquire the coefficients that are unknown for the spacetime domain. In addition, the Neumann boundary conditions are also considered in this study.

$$
h_n(r,\theta,t) = \frac{\partial h(r,\theta,t)}{\partial n} = \nabla h(r,\theta,t) \cdot \vec{n},
\tag{19}
$$

where $\vec{n} = (n_x, n_y)$ denotes the outward normal vector, $n_x$ and $n_y$ are the outward normal direction of the $x$ and $y$ axis, respectively. Adopting the chain rule, we may yield the formulations of $h_n$, $h_x$, and $h_y$, as listed in Appendix B.

### 3.2. The Spacetime Collocation Scheme

Instead of utilizing the original Euclidean space, we adopt a spacetime collocation scheme based on the Minkowski spacetime to perform the transient modeling of this problem. A two-dimensional transient nonlinear moving boundary problem is two-dimensional in space as well as one-dimensional in time, as displayed in Figure 1a. The spacetime region then becomes a domain in three dimensions, as shown in Figure 1b. To calculate the polar angle as well as the radius, we placed the source point as a reference point within the domain, as shown in Figure 1b. As a result, both the initial and boundary condition can be provided on the boundary of spacetime. Since the final time boundary data are unknown, the spacetime collocation scheme transforms a transient nonlinear moving boundary problem into an inverse BVP.

### 3.3. The Iterative Scheme for Modeling Transient Moving Boundary

For each collocation point on the moving surface, the total head is expressed as

$$
h_\varphi(r_j,\theta_j,t_j) = Y_j + \frac{p_j}{\gamma},
\tag{20}
$$

where $Y_j$ is the height above the sea level, $\gamma$ denotes the unit weight of water, $p_j$ denotes the pore water pressure, $h_\varphi(r_j,\theta_j,t_j)$ is the total head, and the subscript $j$ denotes the index of the points on the transient moving boundary to be renewed. The over-specified moving boundary conditions, including the no-flux and the zero pressure head, are described as

$$
\frac{\partial h_\varphi(r_j,\theta_j,t_j)}{\partial n} = 0,\ h_\varphi(r_j,\theta_j,t_j) = Y_j.
\tag{21}
$$

For each collocation point on the seepage face, the total head is expressed as

$$h_\varphi(r_j, \theta_j, t_j) = Y_j. \tag{22}$$

As demonstrated in Equations (16) and (A11), the complete mathematical expressions of the Dirichlet and Neumann boundary conditions have been derived. Applying the Dirichlet and Neumann boundary values for boundary points on the moving surface may acquire

$$h_\varphi(r_j, \theta_j, t_j) \approx \sum_{q=1}^{v} \sum_{p=1}^{v} \mathbf{y}_{qp} \mathbf{H}_{qp}(r_j, \theta_j, t_j), \tag{23}$$

$$\frac{\partial h_\varphi(r_j, \theta_j, t_j)}{\partial n} \approx \sum_{q=1}^{v} \sum_{p=1}^{v} \mathbf{y}_{qp} \frac{\partial \mathbf{H}_{qp}(r_j, \theta_j, t_j)}{\partial n}. \tag{24}$$

From the above equations, the given boundary data are over-specified on the moving boundary. On the moving boundary, the location of the moving boundary is unknown. It can be referred to as the inverse geometric problem. For example, considering the no-flux and the zero pressure head boundary conditions, the unknowns are the coordinates of collocation points. From Equations (23) and (24), it is found that we may solve a nonlinear system of equations to obtain the coordinates of collocation points for the given time. The moving boundary problem may, therefore, exhibit the nonlinear characteristic. The inverse geometric problems are usually difficult to deal with because of the nonlinearity. For solving the inverse geometric problem, such as the moving surface flow problem, the iterative scheme is required. Previous studies have found it difficult to calculate the Jacobian matrix using Newton's method. Thus, the Picard iterative method is used in this study [5]. The Picard iteration first begins from the initial guess of the location for the moving boundary. The iteration may be achieved by applying Equations (23) and (24).

$$\sum_{q=1}^{v} \sum_{p=1}^{v} \mathbf{y}_{qp}^{i} \mathbf{H}_{qp}(r_j^i, \theta_j^i, t_j^i) = h^i(r_j, \theta_j, t_j), \tag{25}$$

$$\sum_{q=1}^{v} \sum_{p=1}^{v} \mathbf{y}_{qp}^{i} \frac{\partial \mathbf{H}_{qp}(r_j^i, \theta_j^i, t_j^i)}{\partial n} = \frac{\partial h^i(r_j, \theta_j, t_j)}{\partial n}, \tag{26}$$

where $h^i(r_j, \theta_j, t_j) = Y_j^i$ and the superscripts $i$ is the number of iteration steps. The iterative equation is depicted as

$$h^i(r_j, \theta_j, t_j) = h^{i-1}(r_j, \theta_j, t_j) + \varepsilon(h^i(r_j, \theta_j, t_j) - h^{i-1}(r_j, \theta_j, t_j)), \tag{27}$$

where $h^i(r_j, \theta_j, t_j)$ is the total head to be renewed, and $\varepsilon$ is the parameter of under-relaxation. The $\varepsilon$ value is in the range of zero to one. The numerical procedure of the iteration starts by giving an initial value for the nonlinear moving boundary and ends while the stopping condition is achieved.

$$\frac{\sqrt{\sum_{j=1}^{J}\left[h^i(r_j, \theta_j, t_j) - h^{i-1}(r_j, \theta_j, t_j)\right]^2}}{\sqrt{\sum_{j=1}^{J}\left[h^{i-1}(r_j, \theta_j, t_j)\right]^2}} \leq \omega, \tag{28}$$

where $\omega$ is the stopping criteria, and $J$ is the collocation point number on the moving boundary. In this study, we consider the stopping criteria to be $\omega = 10^{-4}$.

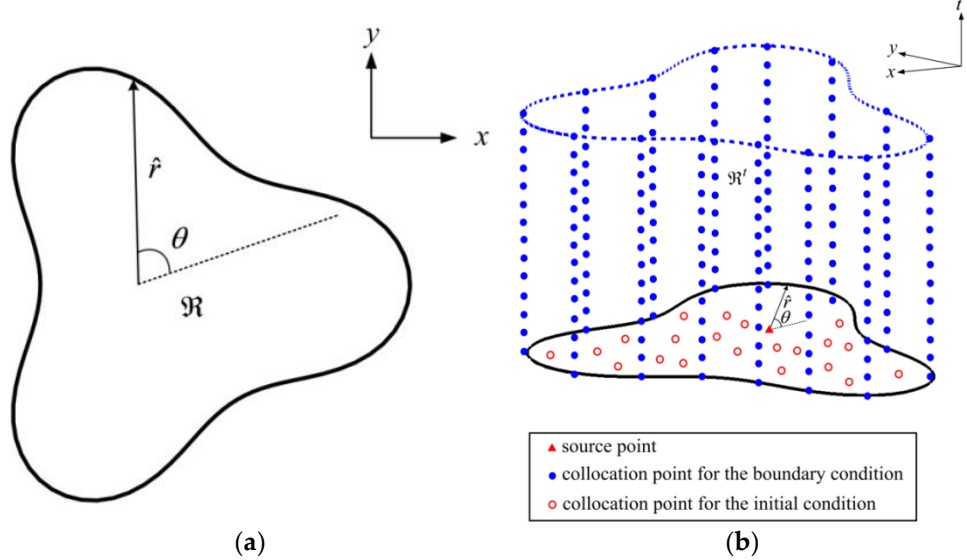

**Figure 1.** The original domain and the spacetime collocation scheme: (**a**) the original two-dimensional space; (**b**) the two-dimensional spacetime collocation.

## 4. Numerical Examples

### 4.1. Numerical Example 1

An example of the shape of a Cassini oval, as demonstrated in Figure 1a, is expressed as

$$\mathfrak{R} \in \left\{ (x, y) \big| x = \hat{r} \cos \theta, y = \hat{r} \sin \theta \right\}, \tag{29}$$

where $\hat{r}(\theta) = \sqrt[3]{\cos(3\theta) + \sqrt{2 - \sin(3\theta)^2}}$, $0 \le \theta \le 2\pi$. The governing equation can be expressed as given in Equation (1). The harmonic data at the initial time is assumed as

$$h(x, y, \hat{t} = 0) = x^2 + y^2. \tag{30}$$

The Neumann data are applied on the domain boundary $\Gamma$, as displayed in Figure 1b. The Neumann boundary condition is considered as

$$\frac{\partial h(x, y, \hat{t})}{\partial n} = 2x \cdot n_x + 2y \cdot n_y \text{ on } \Gamma. \tag{31}$$

The following exact solution is adopted to validate the proposed method.

$$h(x, y, \hat{t}) = x^2 + y^2 + \frac{1}{4}\frac{T}{S}\hat{t} \tag{32}$$

In this example, the storage coefficient $S$ is $10^{-4}$, the transmissibility coefficient $T$ is $10^{-5}$ m$^2$/s, and final elapsed time $\hat{t}_f$ is 3 s. This example demonstrates space collocation points in two dimensions and time collocation points in one dimension. The spacetime collocation points can be regarded as a spacetime domain in three dimensions, as shown in Figure 1b. Due to the inaccessible final time boundary data, the two-dimensional initial value problem is transformed into a three-dimensional inverse BVP. The initial, as well as boundary data, are assigned on the circumferential and bottom sides of the spacetime domain in three dimensions, respectively. In this example, the source point is collocated on the origin as a reference point for calculating the polar angle and radius, as shown in Figure 1b.

The accuracy of the solution of our approach may be affected by the boundary collocation points number as well as the order of the Trefftz basis functions. A sensitivity analysis of the boundary collocation points number, and the order of the Trefftz basis functions is then carried out. To verify the stability of the proposed method, the accuracy of the numerical solution is measured by the following maximum absolute error (MAE).

$$\mathrm{MAE} = \max|u_E - u_N|, \tag{33}$$

where $u_E$ is the exact solution, and $u_N$ is the numerical solution.

Figure 2a demonstrates the relationship between the MAE and the order of the basis functions. It seems that accurate solutions are yielded when the order of the basis functions is greater than 10. Figure 2b depicts the number of boundary collocation points versus the MAE. It seems that accurate solutions may be achieved when the boundary collocation points number is greater than 700. Hence, the order of the basis functions and the boundary collocation points number are considered to be 11 and 918, respectively.

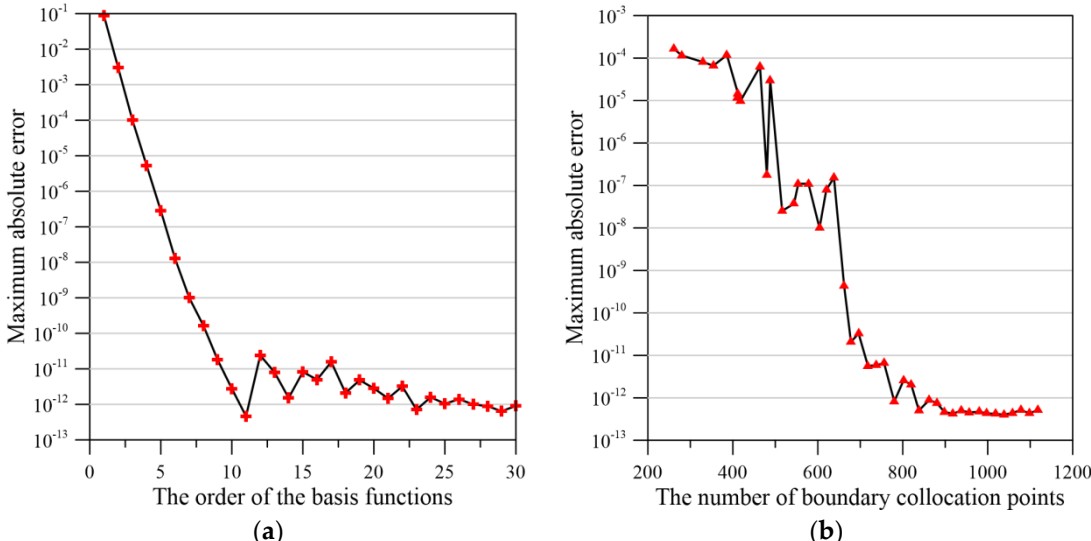

**Figure 2.** The sensitivity analysis: (**a**) accuracy for the order of the basis functions versus the maximum absolute error; (**b**) accuracy for the boundary collocation points number versus the maximum absolute error.

An example of a two-dimensional transient subsurface flow problem with harmonic initial and boundary conditions is then carried out to verify the computed result. To yield the computed total head and examine the accuracy of the proposed method, 3158 inner collocation points are uniformly collocated within the domain. The profiles of the computed total head are then chosen to compare with the analytical solution. Figure 3 depicts the exact solution as well as the field solution of the total head from the proposed method. It seems that by utilizing our method, the computed results are entirely consistent with the analytical solution. Figure 4 depicts the MAE of our method at different times. The MAE of our method is in the order of $10^{-13}$, as displayed in Figure 4. It is clear that our method may yield accurate results.

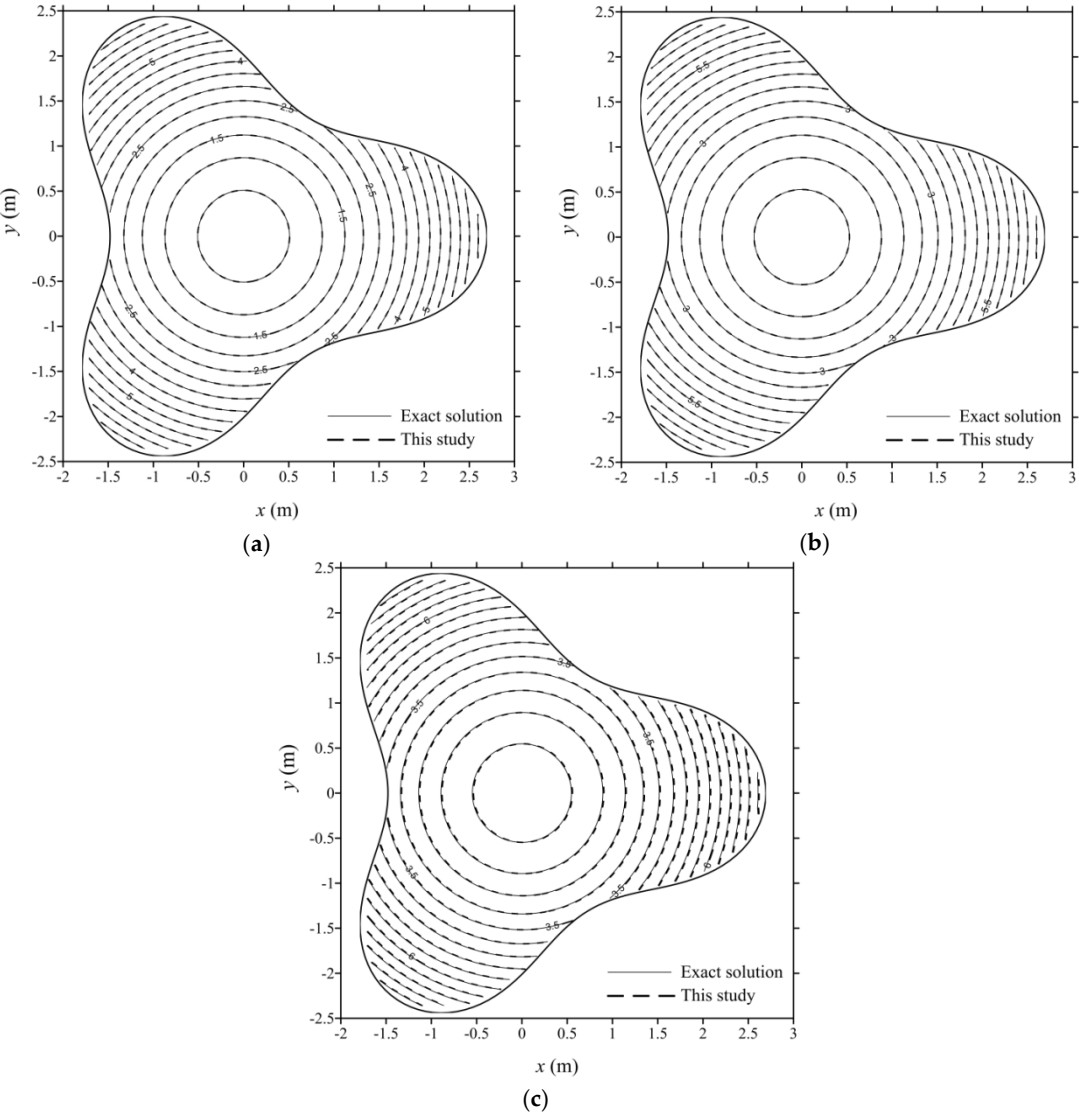

**Figure 3.** Comparison of the solutions with the exact solution: (**a**) $\hat{t} = 0.6$ s; (**b**) $\hat{t} = 1.8$ s; (**c**) $\hat{t} = 3$ s.

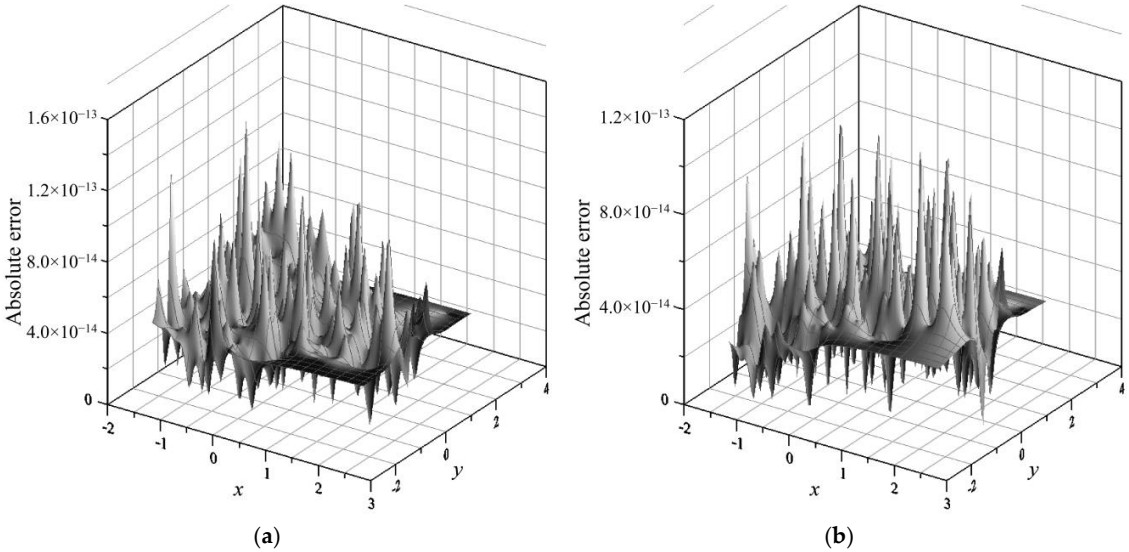

**Figure 4.** *Cont.*

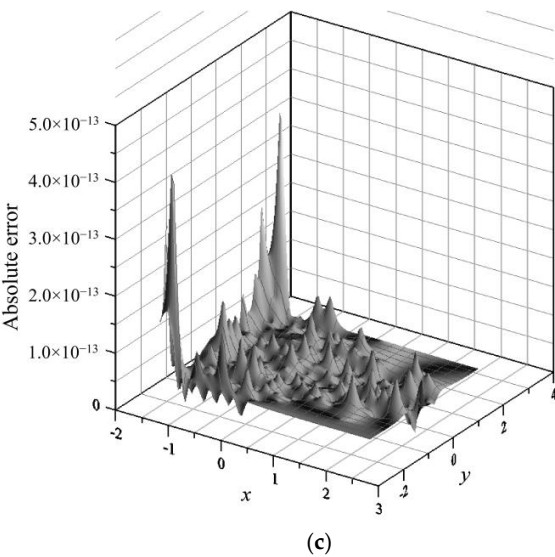

(c)

**Figure 4.** The absolute error at different times: (**a**) $\hat{t} = 0.6$ s; (**b**) $\hat{t} = 1.8$ s; (**c**) $\hat{t} = 3$ s.

### 4.2. Numerical Example 2

This example is the forward and backward analyses of a two-dimensional transient problem. The governing equation is expressed in Equation (1). The non-harmonic initial and boundary data are considered. In this example, we assume the final elapsed time to be 1 min, the storage coefficient to be $10^{-4}$, the transmissibility coefficient to be $10^{-6}$ m$^2$/s, the length to be 10 m, and the width to be 6 m. We apply the non-harmonic initial and boundary data as

$$h(x, y, \hat{t} = 0) = 100 \sin x \sin y, \tag{34}$$

$$h(x, y, \hat{t}) = 100 e^{-\frac{3T}{S}\hat{t}} \sin x \sin y. \tag{35}$$

Because numerical example 2 may not exist an analytical solution to examine the accuracy, we conduct the forward modeling of the two-dimensional transient problem to compute the final time field solution of total head. The non-harmonic boundary data can be provided on vertical sides of the spacetime domain, as displayed in Figure 5a. A backward analysis using the final time results from the forward analysis, as displayed in Figure 5b, is then carried out to compute the field solution of initial head at the bottom of the spacetime domain. To verify the correctness of the field solution, the assigned non-harmonic initial data is compared with the computed initial head from the backward analysis of this problem.

In this example, there exists one source point collocated on the origin and 600 boundary points uniformly collocated on the boundary. The order of the Trefftz function is 21. To yield the computed total head and examine the accuracy of the proposed method, 1000 inner collocation points are uniformly collocated within the domain. Figure 6 shows the computed initial head with the assigned non-harmonic initial data at different times. It is found that the computed initial data from the backward analysis are consistent with the assigned non-harmonic data at time zero. Moreover, the absolute difference is in the order of $10^{-7}$, as given in Figure 7.

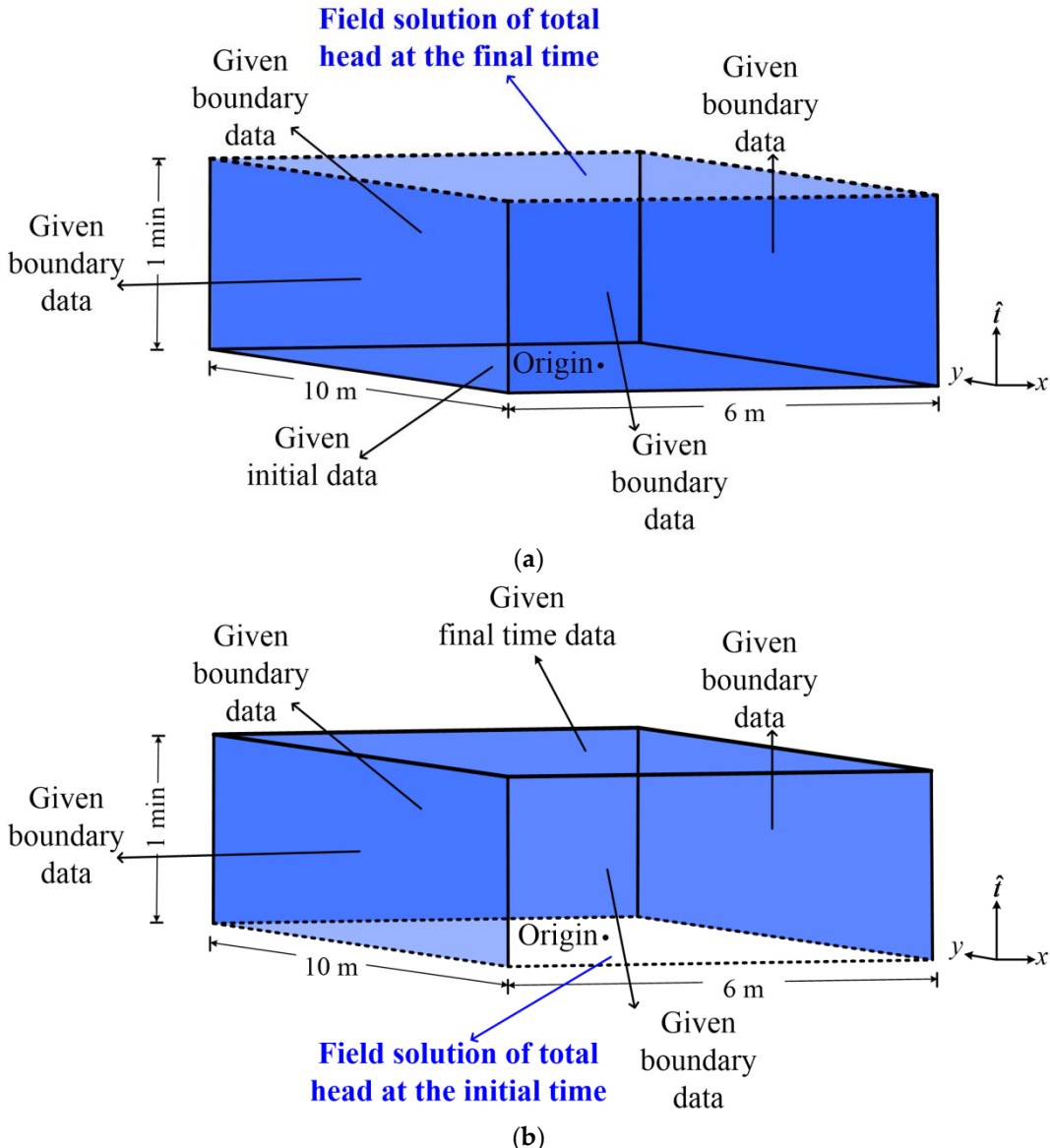

**Figure 5.** Illustration of the forward and backward analyses: (**a**) a forward analysis; (**b**) a backward analysis.

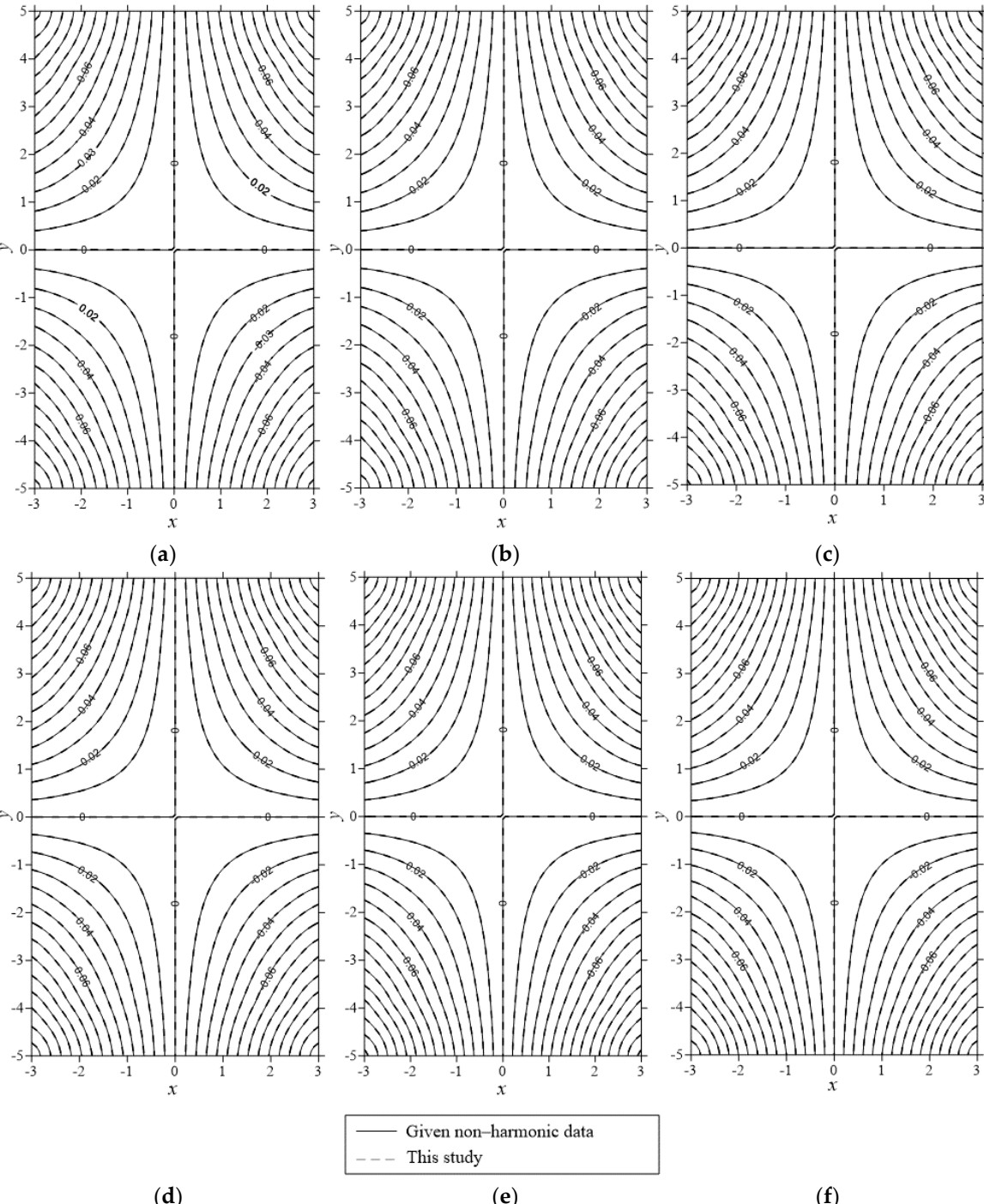

**Figure 6.** Comparison of the numerical solutions with the assigned non-harmonic conditions:
(**a**) $\hat{t} = 1.0$ min; (**b**) $\hat{t} = 0.8$ min; (**c**) $\hat{t} = 0.6$ min; (**d**) $\hat{t} = 0.4$ min; (**e**) $\hat{t} = 0.2$ min; (**f**) $\hat{t} = 0$ min.

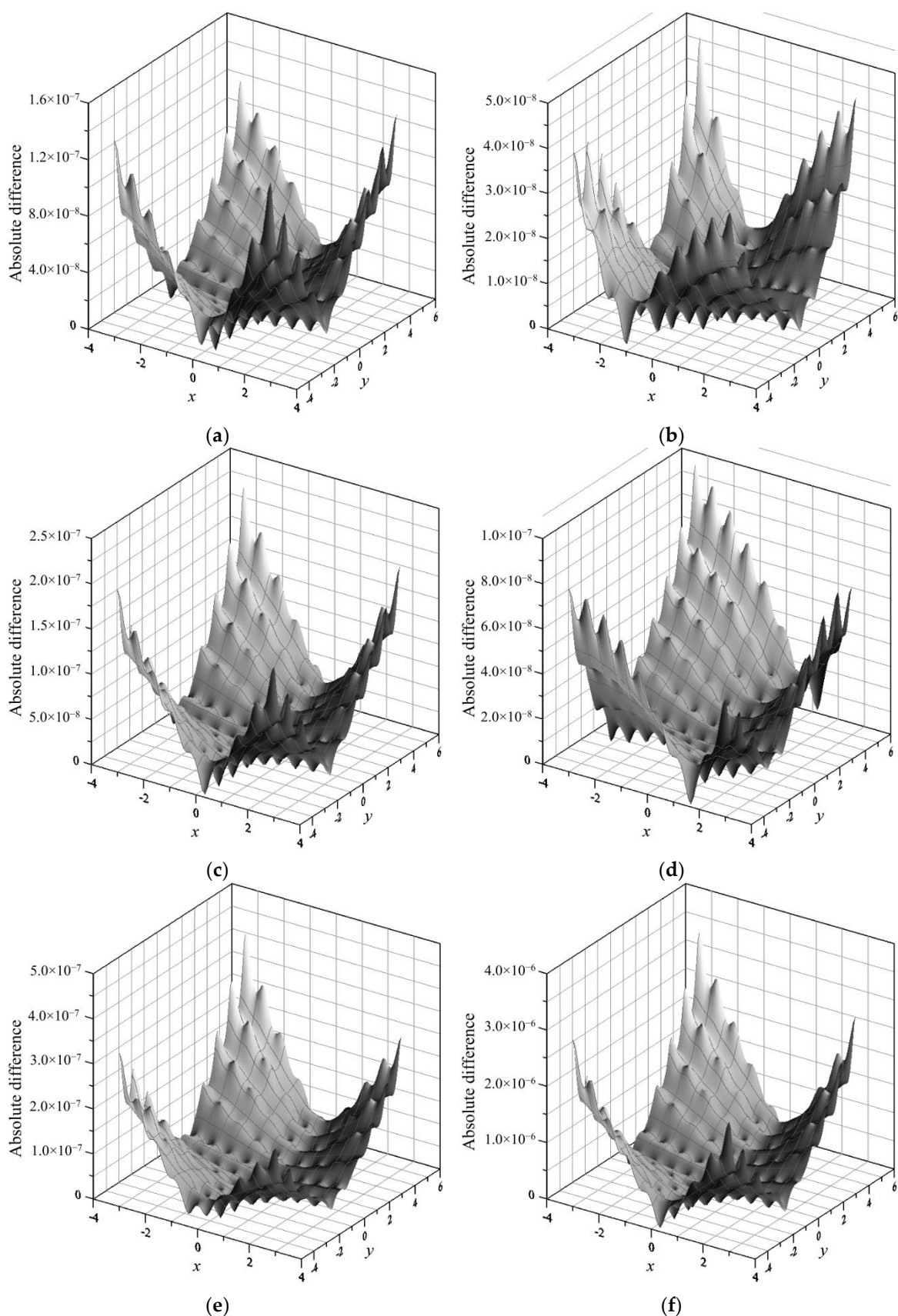

**Figure 7.** The absolute difference at different time: (**a**) $\hat{t} = 1.0$ min; (**b**) $\hat{t} = 0.8$ min; (**c**) $\hat{t} = 0.6$ min; (**d**) $\hat{t} = 0.4$ min; (**e**) $\hat{t} = 0.2$ min; (**f**) $\hat{t} = 0$ min.

### 4.3. Numerical Example 3

The modeling of a two-dimensional transient moving boundary problem through a rectangular dam is considered, as depicted in Figure 8. The objective of this two-dimensional problem is to evaluate the time-dependent location of the phreatic surface. The governing equation is expressed in Equation (1). The rectangular dam, as shown in Figure 8, is constituted of five boundary lines, including $\Gamma_1$, $\Gamma_2$, $\Gamma_3$, $\Gamma_4$, and $\Gamma_5$. We assume the downstream head, upstream head, and the width to be 4, 24, and 16 m, respectively. The initial condition is expressed as

$$h(x, y, \hat{t} = 0) = 24 \text{ m}. \tag{36}$$

The Dirichlet data are applied on the domain boundary, including $\Gamma_2$, $\Gamma_3$, $\Gamma_4$, and $\Gamma_5$, as depicted in Figure 8. The boundary conditions are expressed as

$$h(x, y, \hat{t}) = 4 \text{ m on } \Gamma_2, \tag{37}$$

$$h(x, y, \hat{t}) = 24 \text{ m on } \Gamma_5, \tag{38}$$

$$h(x, y, \hat{t}) = y \text{ m on } \Gamma_3 \text{ and } \Gamma_4. \tag{39}$$

On $\Gamma_1$ and $\Gamma_4$, no-flow Neumann boundary data can be given as

$$\frac{\partial h(x, y, \hat{t})}{\partial n} = 0 \text{ on } \Gamma_1 \text{ and } \Gamma_4. \tag{40}$$

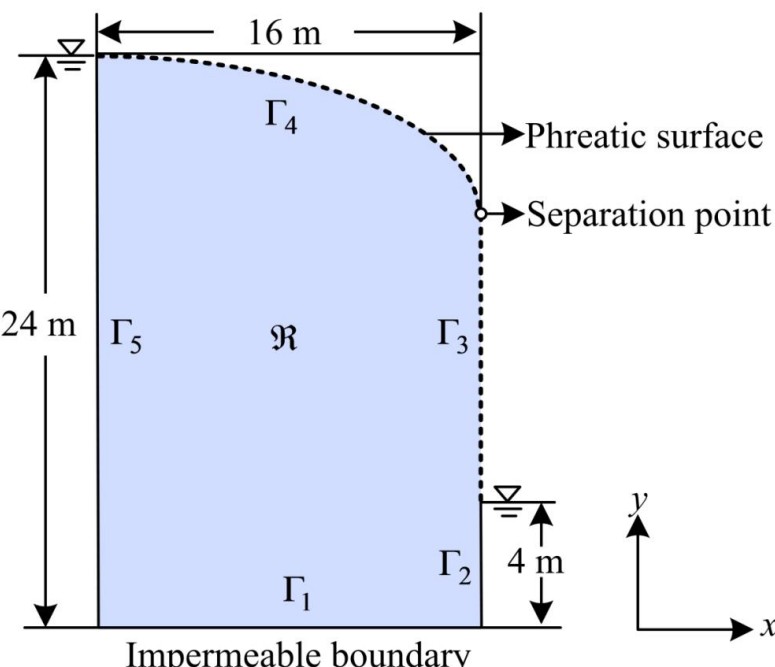

**Figure 8.** Boundary conditions of a two-dimensional moving boundary problem.

In this example, we assume that the final elapsed time is 700 days, storage coefficient is $10^{-3}$ and transmissibility coefficient is $10^{-6}$ m$^2$/s. There exists one source point, where the location of the source point is (0,12). The order of the Trefftz basis functions and boundary collocation points number on a nonlinear moving boundary are 7 and 25,600, respectively. Since the numerical procedure for evaluating the location of the phreatic surface is considered as an inverse problem, the location of the separation point has to be investigated. To obtain the results, the number of iterations is 43 using

the proposed method. Figure 9 shows the numerical solutions of the nonlinear moving boundary at different times. It seems that the transient nonlinear moving boundary can be obtained by utilizing our method.

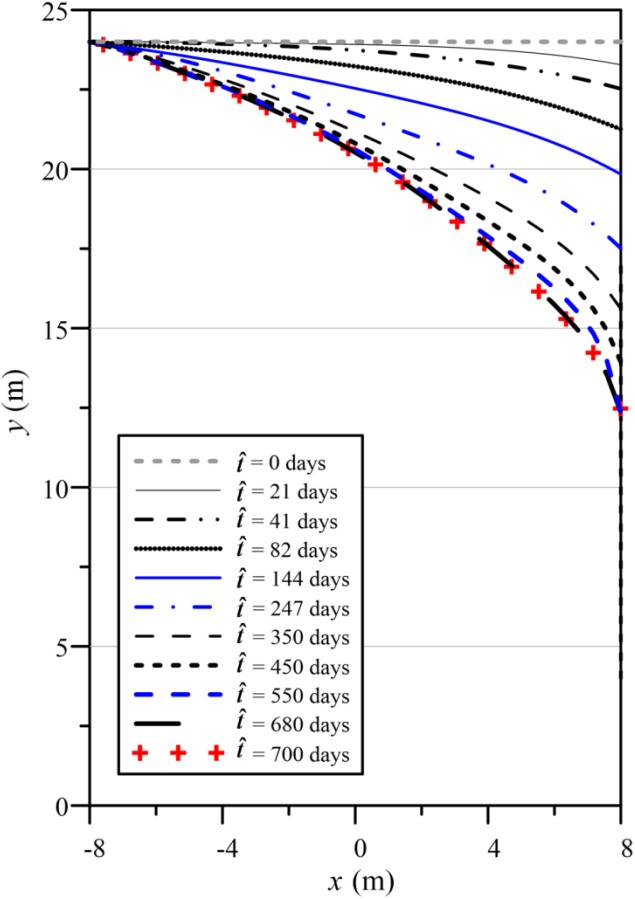

**Figure 9.** Computed moving boundary at different times.

Since several numerical approaches have been applied to solve this problem, we further compare the final time solutions of our method with those of Aitchison (1972) [31], Chen et al. (2007) [9], and Ku et al. (2019) [23], as depicted in Figure 10. It is found that the location of the nonlinear moving boundary using the proposed method agrees very well with the results from the previous studies at the final steady-state time.

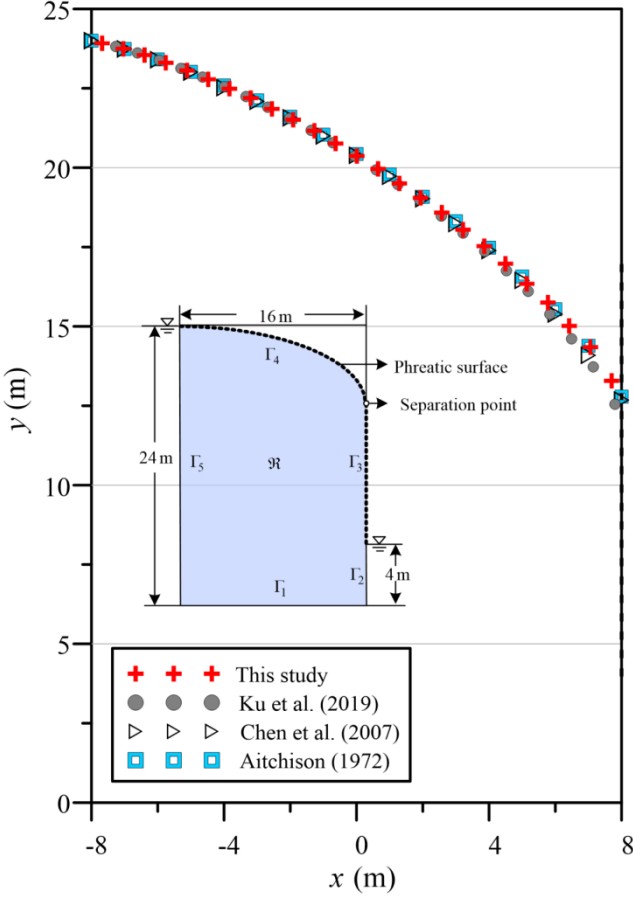

**Figure 10.** Comparison of the computed moving boundary at the final steady-state time.

*4.4. Numerical Example 4*

The final example is the modeling of a two-dimensional transient moving boundary problem through a trapezoidal dam, as displayed in Figure 11. The objective of this example is to evaluate the position of the transient moving boundary. The governing equation is expressed in Equation (1). The initial data is given as

$$h(x, y, \hat{t} = 0) = 4 \text{ m.} \tag{41}$$

The Dirichlet data are applied on the domain boundary, including $\Gamma_2$, $\Gamma_3$, $\Gamma_4$, and $\Gamma_5$, as depicted in Figure 11. The boundary conditions are described as

$$h(x, y, \hat{t}) = 3 \text{ m on } \Gamma_2, \tag{42}$$

$$h(x, y, \hat{t}) = 4 \text{ m on } \Gamma_5, \tag{43}$$

$$h(x, y, \hat{t}) = y \text{ m on } \Gamma_3 \text{ and } \Gamma_4. \tag{44}$$

On $\Gamma_1$ and $\Gamma_4$, the no-flow Neumann boundary data are given as

$$\frac{\partial h(x, y, \hat{t})}{\partial n} = 0 \text{ on } \Gamma_1 \text{ and } \Gamma_4. \tag{45}$$

In numerical example 4, the final elapsed time is 30 min, storage coefficient is $10^{-4}$, and transmissibility coefficient is $10^{-6}$ m$^2$/s. The order of the Trefftz basis functions and number of boundary collocation points on the transient moving boundary are set to be 10 and 9663, respectively. There exists one source point, where the location of the source point is (1.2,2). Figure 12 shows the

proposed spacetime collocation scheme of the two-dimensional transient moving boundary flow through a trapezoidal dam.

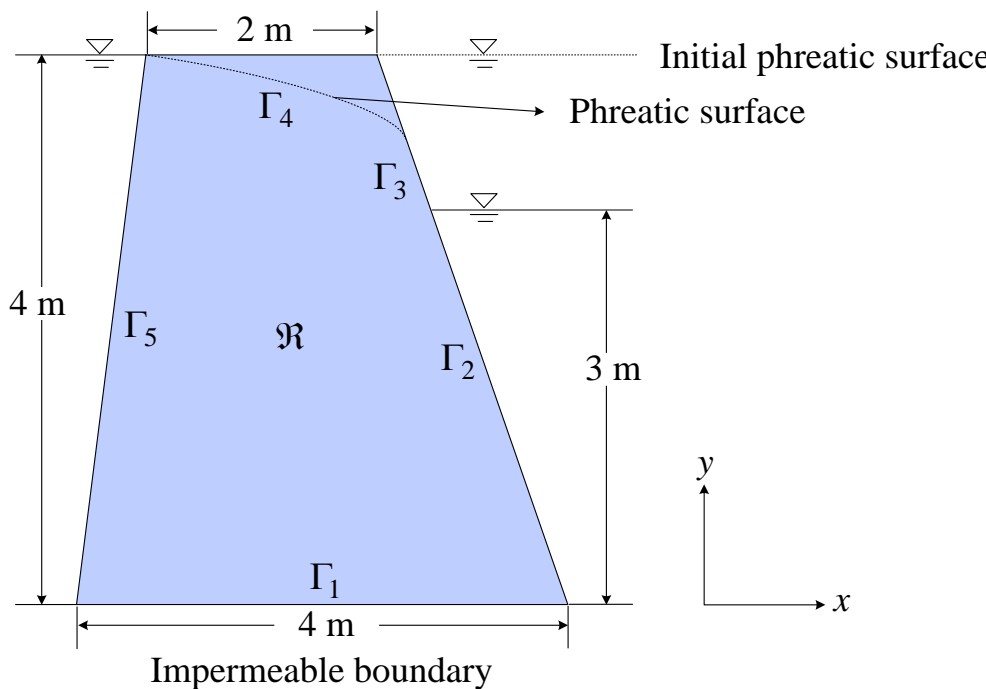

**Figure 11.** The domain of a two-dimensional transient moving boundary problem.

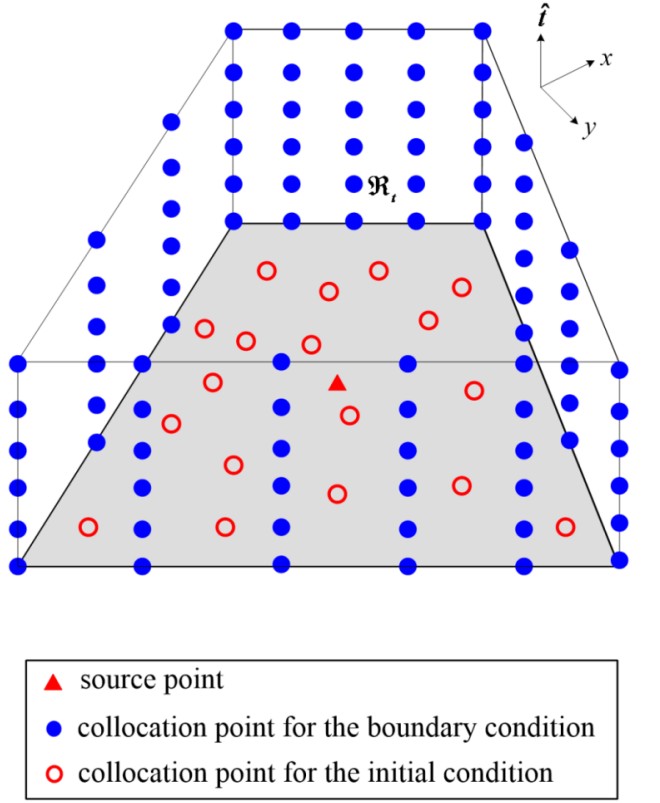

**Figure 12.** The collocation scheme of the two-dimensional transient moving boundary flow through a trapezoidal dam.

The profiles of the field solutions at different simulation times are chosen to clearly view the computed transient moving boundary. Figure 13 shows the numerical solutions of the transient moving boundary. The location of the moving surface at the final time from the proposed method is further compared with the steady-state solution by adopting the MFS to examine the accuracy of the proposed approach, as depicted in Figure 13. It is found that the location of the nonlinear moving surface using the proposed method agrees well with the steady-state solution by using the MFS.

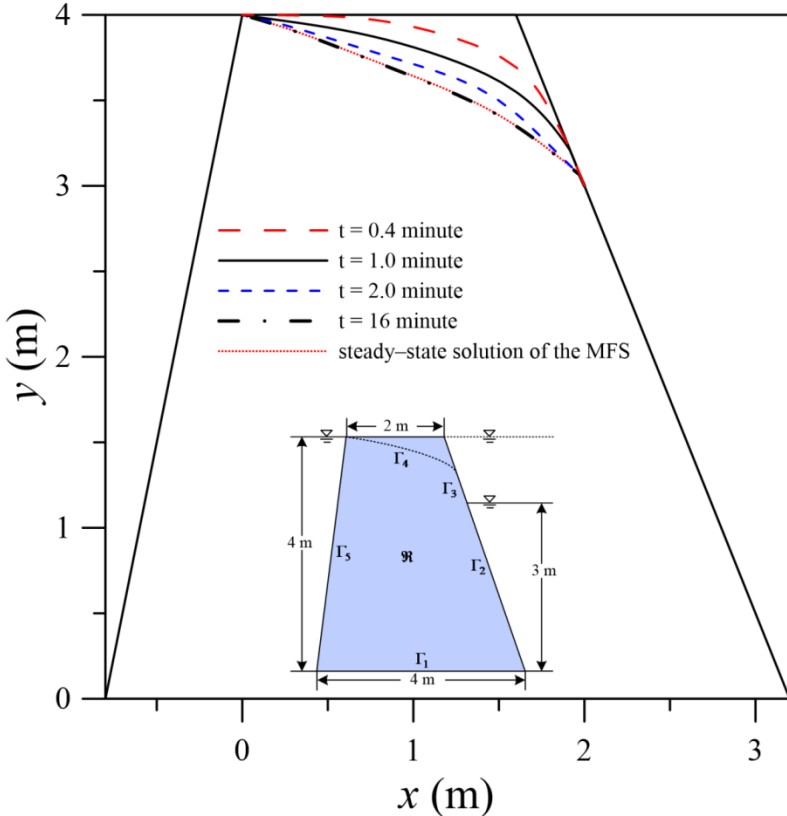

**Figure 13.** The computed nonlinear moving boundary results at different times.

## 5. Conclusions

This study is rooted in the Trefftz method and gives a promising numerical solution for the transient nonlinear moving boundary problems. To verify the proposed spacetime collocation scheme using Trefftz functions, we carried out several numerical problems. The key contributions of this study are as follows.

Previous studies have demonstrated that the engineering application of the Trefftz method with complete Trefftz functions for dealing with transient problems is still hardly found, where solving transient moving boundary problems using the Trefftz method rarely even exists. In this study, a pioneering attempt reveals that the transient nonlinear moving boundary problems governed by the two-dimensional diffusion equation are solved using the spacetime collocation scheme with complete Trefftz basis functions.

The significance of the proposed method rooted in the conventional Trefftz method is that the collocation points in our method are placed in the Minkowski spacetime rather than the Euclidean space. As a result, we may construct a spacetime domain in three dimensions, where both the boundary and initial data are given on the boundary of spacetime, which can be regarded as a BVP. Accordingly, the transient nonlinear moving boundary problems can be easily solved.

Because our method is a boundary-type meshless approach, the domain boundary has to be discretized by the boundary points. It depicts the simplicity of using the proposed method for dealing with problems of the transient moving boundary during the iterative process for searching the location of the free surface.

**Author Contributions:** Conceptualization, C.-Y.K.; methodology, C.-Y.L.; validation, C.-Y.L.; visualization, J.-E.X.; writing—original draft preparation, C.-Y.L.; writing—review and editing, C.-Y.K. and C.-M.F. supervision, W.Y.

**Funding:** This research was partially funded by the Ministry of Science and Technology of the Republic of China under grant MOST 108-2621-M-019-008.

**Acknowledgments:** We sincerely thank the Ministry of Science and Technology for the generous funding. The first author is also grateful to his former graduate student, Feng Kao, for her assistance of this article.

**Conflicts of Interest:** The authors declare no conflict of interest.

## Appendix A

The formulation of Trefftz functions for transient moving boundary problems are expressed in the following description.

(I) $\lambda = 0$ and $\chi = 0$

Assuming $\lambda = 0$ and $\chi = 0$, we obtain

$$\begin{cases} \Omega(t) = A_6, \\ R(r) = A_7 \ln r + A_8, \\ W(\theta) = A_9 \theta + A_{10}, \end{cases} \tag{A1}$$

where $A_6$, $A_7$, $A_8$, $A_9$, and $A_{10}$ are constants. Applying the boundary conditions of $W(r, 0, t) = W(r, 2\pi, t)$, we obtain that $A_9 = 0$. Inserting Equation (A1) into Equation (5), we find

$$h(r, \theta, t) = \overline{A}_5 \ln r + \overline{A}_6, \tag{A2}$$

where $\overline{A}_5$ and $\overline{A}_6$ are constant.

(II) $\lambda = p^2$ and $\chi = q^2$

Assuming $\lambda = p^2$ and $\chi = q^2$, we obtain

$$\begin{cases} \Omega(t) = A_{11} e^{p^2 t}, \\ R(r) = A_{12} I_q(pr) + A_{13} K_q(pr), \\ W(\theta) = A_{14} \cos(q\theta) + A_{15} \sin(q\theta), \end{cases} \tag{A3}$$

where $A_{11}, A_{12}, A_{13}, A_{14},$ and $A_{15}$ are constants, $I_q$ denotes the modified Bessel function of the first kind of q order, and $K_q$ denotes the modified Bessel function of the second kind of q order. Substituting Equation (A3) into Equation (5), we have

$$\begin{aligned} h(r, \theta, t) = &\overline{A}_7 e^{p^2 t} I_q(pr) \cos(q\theta) + \overline{A}_8 e^{p^2 t} I_q(pr) \sin(q\theta) \\ &+ \overline{A}_9 e^{p^2 t} K_q(pr) \cos(q\theta) + \overline{A}_{10} e^{p^2 t} K_q(pr) \sin(q\theta), \end{aligned} \tag{A4}$$

where $\overline{A}_7, \overline{A}_8, \overline{A}_9,$ and $\overline{A}_{10}$ are constant.

(III) $\lambda = p^2$ and $\chi = 0$

Assuming $\lambda = p^2$ and $\chi = 0$, we obtain

$$\begin{cases} \Omega(t) = A_{16}e^{p^2 t}, \\ R(r) = A_{17}I_0(pr) + A_{18}K_0(pr) + A_{19}, \\ W(\theta) = A_{20}\theta + A_{21}, \end{cases} \tag{A5}$$

where $A_{16}$, $A_{17}$, $A_{18}$, $A_{19}$, $A_{20}$, and $A_{21}$ are constants, $I_0$ denotes the modified Bessel function of the first kind of zero order, and $K_0$ denotes the modified Bessel function of the second kind of zero order. Applying the boundary conditions of $W(r, 0, t) = W(r, 2\pi, t)$ may obtain $A_{20} = 0$. Inserting Equation (A5) into Equation (5) yields

$$h(r, \theta, t) = \overline{A}_{11}e^{p^2 t}I_0(pr) + \overline{A}_{12}e^{p^2 t}K_0(pr) + \overline{A}_{13}, \tag{A6}$$

where $\overline{A}_{11}$, $\overline{A}_{12}$, and $\overline{A}_{13}$ are constant.

(IV) $\lambda = -p^2$ and $\chi = q^2$

Assuming $\lambda = -p^2$ and $\chi = q^2$, we obtain

$$\begin{cases} T(t) = A_{22}e^{-p^2 t}, \\ R(r) = A_{23}J_q(pr) + A_{24}Y_q(pr), \\ W(\theta) = A_{25}\cos(q\theta) + A_{26}\sin(q\theta), \end{cases} \tag{A7}$$

where $A_{22}$, $A_{23}$, $A_{24}$, $A_{25}$, and $A_{26}$ are constants, $J_q$ denotes the Bessel function of the first kind of $q$ order, $Y_q$ denotes the Bessel function of the second kind of $q$ order. Substituting Equation (A7) into Equation (5), we may have

$$\begin{aligned} h(r, \theta, t) = &\overline{A}_{14}e^{-p^2 t}J_q(pr)\cos(q\theta) + \overline{A}_{15}e^{-p^2 t}J_q(pr)\sin(q\theta) \\ &+\overline{A}_{16}e^{-p^2 t}Y_q(pr)\cos(q\theta) + \overline{A}_{17}e^{-p^2 t}Y_q(pr)\sin(q\theta), \end{aligned} \tag{A8}$$

where $\overline{A}_{14}$, $\overline{A}_{15}$, $\overline{A}_{16}$, and $\overline{A}_{17}$ are constant.

(V) $\lambda = -p^2$ and $\chi = 0$

Assuming $\lambda = -p^2$ and $\chi = 0$, we obtain

$$\begin{cases} \Omega(t) = A_{27}e^{-p^2 t}, \\ R(r) = A_{28}J_0(pr) + A_{29}Y_0(pr) + A_{30}, \\ W(\theta) = A_{31}\theta + A_{32}, \end{cases} \tag{A9}$$

where $A_{27}$, $A_{28}$, $A_{29}$, $A_{30}$, $A_{31}$, and $A_{32}$ are constants, $J_0$ denotes the Bessel function of the first kind of zero order, and $Y_0$ denotes the Bessel function of the second kind of zero order. Using the boundary conditions of $W(r, 0, t) = W(r, 2\pi, t)$, we obtain that $A_{31} = 0$. Substituting Equation (A9) into Equation (5), we find

$$h(r, \theta, t) = \overline{A}_{18}e^{-p^2 t}J_0(pr) + \overline{A}_{19}e^{-p^2 t}Y_0(pr) + \overline{A}_{20}, \tag{A10}$$

where $\overline{A}_{18}$, $\overline{A}_{19}$, and $\overline{A}_{20}$ are constant.

The transient solutions are described by the principle of linear superposition utilizing the Trefftz functions. The Trefftz basis for transient moving boundary problems consists of a series of linearly independent vectors, including 18 functions, as listed in the following table.

In Table A1, $I_0$ and $I_q$ denote the modified Bessel functions of the first kind of zero order and of $q$ order, respectively. $J_0$ and $J_q$ denote the Bessel functions of the first kind of zero order and of $q$ order, respectively. $K_0$ and $K_q$ denote the modified Bessel functions of the second kind of zero order and of $q$

order, respectively. $Y_0$ and $Y_q$ denote the Bessel functions of the second kind of zero order and of $q$ order, respectively.

**Table A1.** The Trefftz basis for transient moving boundary problems.

| $\overline{T}_1$ | $1$ | $\overline{T}_2$ | $r^q \cos(q\theta)$ |
|---|---|---|---|
| $\overline{T}_3$ | $r^q \sin(q\theta)$ | $\overline{T}_4$ | $r^{-q} \cos(q\theta)$ |
| $\overline{T}_5$ | $r^{-q} \sin(q\theta)$ | $\overline{T}_6$ | $\ln r$ |
| $\overline{T}_7$ | $e^{p^2 t} I_q(pr) \cos(q\theta)$ | $\overline{T}_8$ | $e^{p^2 t} I_q(pr) \sin(q\theta)$ |
| $\overline{T}_9$ | $e^{p^2 t} K_q(pr) \cos(q\theta)$ | $\overline{T}_{10}$ | $e^{p^2 t} K_q(pr) \sin(q\theta)$ |
| $\overline{T}_{11}$ | $e^{p^2 t} I_0(pr)$ | $\overline{T}_{12}$ | $e^{p^2 t} K_0(pr)$ |
| $\overline{T}_{13}$ | $e^{-p^2 t} J_q(pr) \cos(q\theta)$ | $\overline{T}_{14}$ | $e^{-p^2 t} J_q(pr) \sin(q\theta)$ |
| $\overline{T}_{15}$ | $e^{-p^2 t} Y_q(pr) \cos(q\theta)$ | $\overline{T}_{16}$ | $e^{-p^2 t} Y_q(pr) \sin(q\theta)$ |
| $\overline{T}_{17}$ | $e^{-p^2 t} J_0(pr)$ | $\overline{T}_{18}$ | $e^{-p^2 t} Y_0(pr)$ |

**Appendix B**

To formulate the complete expressions of $h_n$, $h_x$, and $h_y$, the chain rule is utilized. The Neumann boundary data of $h_n$, $h_x$, and $h_y$ are expressed as follows,

$$\frac{\partial h}{\partial n} = \frac{\partial h}{\partial x} n_x + \frac{\partial h}{\partial y} n_y, \tag{A11}$$

$$\frac{\partial h}{\partial x} = \frac{\partial h}{\partial r}\frac{\partial r}{\partial x} + \frac{\partial h}{\partial \theta}\frac{\partial \theta}{\partial x}, \tag{A12}$$

$$\frac{\partial h}{\partial y} = \frac{\partial h}{\partial r}\frac{\partial r}{\partial y} + \frac{\partial h}{\partial \theta}\frac{\partial \theta}{\partial y}. \tag{A13}$$

The formulations of $h_n$, $h_x$, and $h_y$ in the polar coordinates may be derived by using a series of mathematical formulations as follows.

$$\frac{\partial h}{\partial r} = \sum_{p=1}^{v}\sum_{q=1}^{v}\begin{bmatrix} \overline{C_1}qr^{q-1}\cos(q\theta) + \overline{C_2}qr^{q-1}\sin(q\theta) - q\overline{C_3}r^{-q-1}\cos(q\theta) \\ -q\overline{C_4}r^{-q-1}\sin(q\theta) + \overline{C_5}\frac{1}{r} \\ +\overline{C_7}\frac{p}{2}e^{p^2 t}(I_{q+1}(pr) - I_{q-1}(pr))\cos(q\theta) \\ +\overline{C_8}\frac{p}{2}e^{p^2 t}(I_{q+1}(pr) - I_{q-1}(pr))\sin(q\theta) \\ +\overline{C_9}\frac{p}{2}e^{p^2 t}(K_{q+1}(pr) - K_{q-1}(pr))\cos(q\theta) \\ +\overline{C_{10}}\frac{p}{2}e^{p^2 t}(K_{q+1}(pr) - K_{q-1}(pr))\sin(q\theta) \\ +\overline{C_{11}}e^{p^2 t}I_1(pr) + \overline{C_{12}}e^{\alpha p^2 t}K_1(pr) \\ +\overline{C_{13}}\frac{p}{2}e^{-p^2 t}(J_{q-1}(pr) - J_{q+1}(pr))\cos(q\theta) \\ +\overline{C_{14}}\frac{p}{2}e^{-p^2 t}(J_{q-1}(pr) - J_{q+1}(pr))\sin(q\theta) \\ +\overline{C_{15}}\frac{p}{2}e^{-p^2 t}(Y_{q-1}(pr) - Y_{q+1}(pr))\cos(q\theta) \\ +\overline{C_{16}}\frac{p}{2}e^{-p^2 t}(Y_{q-1}(pr) - Y_{q+1}(pr))\sin(q\theta) \\ +\overline{C_{17}}e^{-p^2 t}J_1(pr) - \overline{C_{18}}e^{-p^2 t}Y_1(pr) \end{bmatrix}, \tag{A14}$$

$$\frac{\partial h}{\partial \theta} = \sum_{p=1}^{v} \sum_{q=1}^{v} \begin{bmatrix} \overline{C_1} r^q(-\sin(q\theta)) + \overline{C_2} r^q \cos(q\theta) \\ +\overline{C_3} r^{-q}(-\sin(q\theta)) + \overline{C_4} r^{-q} \cos(q\theta) \\ +\overline{C_7} e^{p^2 t} I_q(pr)(-\sin(q\theta)) + \overline{C_8} e^{p^2 t} I_q(pr) \cos(q\theta) \\ +\overline{C_9} e^{p^2 t} K_q(pr)(-\sin(q\theta)) + \overline{C_{10}} e^{p^2 t} K_q(pr) \cos(q\theta) \\ +\overline{C_{13}} e^{-p^2 t} J_q(pr)(-\sin(q\theta)) + \overline{C_{14}} e^{-p^2 t} J_q(pr) \cos(q\theta) \\ +\overline{C_{15}} e^{-p^2 t} Y_q(pr)(-\sin(q\theta)) + \overline{C_{16}} e^{-p^2 t} Y_q(pr) \cos(q\theta) \end{bmatrix} \tag{A15}$$

$$\frac{\partial r}{\partial x} = \frac{x}{\sqrt{x^2 + y^2}} = \frac{x}{r} = \cos\theta, \tag{A16}$$

$$\frac{\partial r}{\partial y} = \frac{y}{\sqrt{x^2 + y^2}} = \frac{y}{r} = \sin\theta, \tag{A17}$$

$$\frac{\partial \theta}{\partial x} = \frac{-y/x^2}{1 + y^2/x^2} = \frac{-y}{r^2} = -\frac{\sin\theta}{r}, \tag{A18}$$

$$\frac{\partial \theta}{\partial y} = \frac{1/x}{1 + y^2/x^2} = \frac{x}{r^2} = \frac{\cos\theta}{r}. \tag{A19}$$

Substituting the above equations into Equations (A12) and (A13), we may obtain the following equations.

$$\frac{\partial h}{\partial x} = \sum_{p=1}^{v} \sum_{q=1}^{v} \begin{bmatrix} \overline{C_1}(qr^{q-1}\cos(q\theta)\cos(\theta) + r^q(-\sin(q\theta))(-\frac{\sin\theta}{r})) \\ +\overline{C_2}(qr^{q-1}\sin(q\theta)\cos(\theta) + r^q \cos(q\theta)(-\frac{\sin\theta}{r})) \\ -\overline{C_3}(qr^{-q-1}\cos(q\theta)\cos(\theta) + r^{-q}(-\sin(q\theta))(-\frac{\sin\theta}{r})) \\ -\overline{C_4}(qr^{-q-1}\sin(q\theta)\cos(\theta) + r^{-q}\cos(q\theta)(-\frac{\sin\theta}{r})) \\ +\overline{C_5}\frac{1}{r}\cos(\theta) + \overline{C_7}\frac{p}{2}e^{p^2 t}((I_{q+1}(pr) - I_{q-1}(pr))\cos(q\theta)\cos(\theta) \\ +I_q(pr)(-\sin(q\theta))(-\frac{\sin\theta}{r})) \\ +\overline{C_8}e^{p^2 t}(\frac{p}{2}(I_{q+1}(pr) - I_{q-1}(pr))\sin(q\theta)\cos(\theta) \\ +I_q(pr)\cos(q\theta)(-\frac{\sin\theta}{r})) \\ +\overline{C_9}e^{p^2 t}(\frac{p}{2}(K_{q+1}(pr) - K_{q-1}(pr))\cos(q\theta)\cos(\theta) \\ +K_q(pr)(-\sin(q\theta))(-\frac{\sin\theta}{r})) \\ +\overline{C_{10}}e^{p^2 t}(\frac{p}{2}(K_{q+1}(pr) - K_{q-1}(pr))\sin(q\theta)\cos(\theta) \\ +K_q(pr)\cos(q\theta)(-\frac{\sin\theta}{r})) \\ +\overline{C_{11}}e^{p^2 t}(I_1(pr)\cos(\theta)) \\ +\overline{C_{12}}e^{p^2 t}(K_1(pr)\cos(\theta)) \\ +\overline{C_{13}}e^{-p^2 t}(\frac{p}{2}(J_{q-1}(pr) - J_{q+1}(pr))\cos(q\theta)\cos(\theta) \\ +J_q(pr)(-\sin(q\theta))(-\frac{\sin\theta}{r})) \\ +\overline{C_{14}}e^{-p^2 t}(\frac{p}{2}(J_{q-1}(pr) - J_{q+1}(pr))\sin(q\theta)\cos(\theta) \\ +J_q(pr)\cos(q\theta)(-\frac{\sin\theta}{r})) \\ +\overline{C_{15}}e^{-p^2 t}(\frac{p}{2}(Y_{q-1}(pr) - Y_{q+1}(pr))\cos(q\theta)\cos(\theta) \\ +Y_q(pr)(-\sin(q\theta)(-\frac{\sin\theta}{r})) \\ +\overline{C_{16}}e^{-p^2 t}(\frac{p}{2}(Y_{q-1}(pr) - Y_{q+1}(pr))\sin(q\theta)\cos(\theta) \\ +e^{-p^2 t}Y_q(pr)\cos(q\theta)(-\frac{\sin\theta}{r})) \\ +\overline{C_{17}}e^{-p^2 t}J_1(pr)\cos(\theta) \\ -\overline{C_{18}}e^{-\alpha p^2 t}Y_1(pr)\cos(\theta) \end{bmatrix} n_x, \tag{A20}$$

$$
\frac{\partial h}{\partial y} = \sum_{p=1}^{v}\sum_{q=1}^{v}
\begin{bmatrix}
\overline{C_1}\left(qr^{q-1}\cos(q\theta)\sin(\theta)+r^{q}(-\sin(q\theta))\left(\frac{\cos\theta}{r}\right)\right)\\
+\overline{C_2}qr^{q-1}\sin(q\theta)\sin(\theta)+r^{q}\cos(q\theta)\left(\frac{\cos\theta}{r}\right))\\
-\overline{C_3}qr^{-q-1}\cos(q\theta)\sin(\theta)+r^{-q}(-\sin(q\theta))\left(\frac{\cos\theta}{r}\right))\\
-\overline{C_4}qr^{-q-1}\sin(q\theta)\sin(\theta)+r^{-q}\cos(q\theta)\left(\frac{\cos\theta}{r}\right))\\
+\overline{C_5}\frac{1}{r}\sin(\theta)+\overline{C_7}e^{\alpha p^2 t}\left(\frac{p}{2}(I_{q+1}(pr)-I_{q-1}(pr))\cos(q\theta)\sin(\theta)\right.\\
+I_q(pr)(-\sin(q\theta))\left(\frac{\cos\theta}{r}\right))\\
+\overline{C_8}e^{\alpha p^2 t}\left(\frac{p}{2}(I_{q+1}(pr)-I_{q-1}(pr))\sin(q\theta)\sin(\theta)\right.\\
+I_q(pr)\cos(q\theta)\left(\frac{\cos\theta}{r}\right))\\
+\overline{C_9}e^{\alpha p^2 t}\left(\frac{p}{2}(K_{q+1}(pr)-K_{q-1}(pr))\cos(q\theta)\sin(\theta)\right.\\
+K_q(pr)(-\sin(q\theta))\left(\frac{\cos\theta}{r}\right))\\
+\overline{C_{10}}e^{\alpha p^2 t}\left(\frac{p}{2}(K_{q+1}(pr)-K_{q-1}(pr))\sin(q\theta)\sin(\theta)\right.\\
+K_q(pr)\cos(q\theta)\left(\frac{\cos\theta}{r}\right))\\
+\overline{C_{11}}e^{\alpha p^2 t}(I_1(pr)\sin(\theta))\\
+\overline{C_{12}}e^{\alpha p^2 t}(K_1(pr)\sin(\theta))\\
+\overline{C_{13}}e^{-\alpha p^2 t}\left(\frac{p}{2}(J_{q-1}(pr)-J_{q+1}(pr))\cos(q\theta)\sin(\theta)\right.\\
+J_q(pr)(-\sin(q\theta))\left(\frac{\cos\theta}{r}\right))\\
+\overline{C_{14}}e^{-\alpha p^2 t}\left(\frac{p}{2}(J_{q-1}(pr)-J_{q+1}(pr))\sin(q\theta)\sin(\theta)\right.\\
+J_q(pr)\cos(q\theta)\left(\frac{\cos\theta}{r}\right))\\
+\overline{C_{15}}e^{-\alpha p^2 t}\left(\frac{p}{2}(Y_{q-1}(pr)-Y_{q+1}(pr))\cos(q\theta)\sin(\theta)\right.\\
+Y_q(pr)(-\sin(q\theta))\left(\frac{\cos\theta}{r}\right))\\
+\overline{C_{16}}e^{-\alpha p^2 t}\left(\frac{p}{2}(Y_{q-1}(pr)-Y_{q+1}(pr))\sin(q\theta)\sin(\theta)\right.\\
+Y_q(pr)\sin(q\theta)\left(\frac{\cos\theta}{r}\right))\\
+\overline{C_{17}}e^{-\alpha p^2 t}J_1(pr)\sin(\theta)\\
-\overline{C_{18}}e^{-\alpha p^2 t}Y_1(pr)\cos(\theta)
\end{bmatrix}
n_y.
\tag{A21}
$$

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
