# Peer review of "A Spacetime Meshless Method for Modeling Subsurface Flow with a Transient Moving Boundary"

_water, doi:10.3390/w11122595_

Round 1

Reviewer 1 Report

Peer Review Report

Ms. Ref. No.: water-648997

Title: A spacetime meshless method for modeling subsurface flow with transient moving boundary

Authors: Cheng-Yu Ku, Chih-Yu Liu, Jing-En Xiao, Weichung Yeih, Chia-Ming Fan

The paper presents application of the Trefftz method for modeling subsurface flow with transient moving boundary. The authors presented very interesting and worthy of investigations subject in their manuscript. The results are very promising. I recommend the paper for minor revision. The subject of the article is within scope of the journal. The mathematical development is sound. I believe that the authors will find below some suggestions, which will help them to improve their manuscript:

Major comments:

Style of treating equations. Since the whole manuscript should be a text consisting of sentnces, the equation should not exist as something between two sentences, having nothing to do with these sentences – not beeing a part of these sentences. The equations should be the parts of the sentences. To do this, please:

h page 2, lne 71: remove the dot,

page 2, equation (1): add comma after R_t in the end of the formula,

page 2, line 72: please write the word “where” instead of “In the precending equations”, remove also the comma after this term – before R_t,

page 2, line 78: remove the dot in the end of the sentence in this line,

page 3, line 90: remove the dot in the end of the sentence in this line,

page 3, equation (5): remove dot, write comma in this place,

page 3, line 91: please write the word “where” instead of “In the precending equations”, remove also the comma after this term – before \phi,

page 3, line 93: remove the dot in the end of the sentence in this line,

page 3, line 97: remove the dot in the end of the sentence in this line,

page 3, equation (8): remove dot, write comma in this place,

page 3, line 98: please write the word “where” instead of “In the precending equations”, remove also the comma after this term – before R(r),

page 4, line 114: remove the dot in the end of the sentence in this line,

page 4, line 117: remove the dot in the end of the sentence in this line,

page 4, line 126: remove the dot in the end of the sentence in this line,

page 5, line 129: remove the dot in the end of the sentence in this line,

page 5, equation (18): add comma in the end of the formula,

page 5, line 139: please remove the term: “, ad depicted in Equation (19),” (by the way, the term “depicted in Equation ...” is rather strange – equation is not a picture, please check it in the whole manuscript),

page 5, line 139: remove the dot in the end of the sentence in this line,

page 7, line 176: remove the dot in the end of the sentence in this line,

page 7, equation (27): please write comma intead of dot in the end of the formula in this equation,

page 7, line 179: please write the word “where” instead of “In the precending equations”, remove also the comma after this term – before h^i(...),

page 7, line 182: please remove the term “, as depicted in Equation (28).”, also the dot in the end of this term

page 7, line 191: remove the dot in the end of the sentence in this line,

page 16, line 281: remove the dot in the end of the sentence in this line,

page 21, line 377: please write “as follows” instead of “as following equations” and remove the dot.

page 22, line 380: remove the dot in the end of the sentence in this line,

page 23, line 382: remove the dot in the end of the sentence in this line.

Please check now the whole manuscript in this context.

Page 3, line 92: The authors wrote:”Each function in equation (5) depends only on one of the variables r,\theta and t”. I'm not sure about that, because the function \phi depends on two variables: r and \theta. By the way, in the sentence the word “or” would be better than “and”.

Page 3, line 103: why \lambda and \ksi are constant? Maybe it is your assumtions here. For example, it is easy to imagine that \Omega = 3*t, then \Omega' = 3 and consequently \lambda = 1/t.

The titlles of the figures, tibles, etc. should exist alone without the text of the manuscript. Thus please do not use abbreviations in these titles:

page 8, figure 2,

page 10, figure 4,

page 13, figure 7 (by the way in these figure you does not present the MAE, because you present just absolute value, if I good understand MAE as the maximal value should be one value (the maximal ona) in the whole domain). Furthermore, maybe the absolute error should be also additionally defined.

Please check it in the whole manuscript.

Please give the mathematical formula for the error MAE. How it is defined? In how many test points the error was evaluated? Please write also in each example how many test points were used to evaluate the error.

Minor comments:

Please use commas:

page 1, linees 13-14: before the word “where”,

page 1, line 15: before the word “in which”

page 1, line 18: before the word “which”,

page 1, line 21: before the word “because”,

page 1, line 31: before the word “because”,

page 2, line 57: before the word “where”,

page 2, line 59: before the word: “in which”,

page 2, line 62: before the word: “which”,

page 19, line 310: before the word: “where”,

page 19, line 317: before the word: “where”,

page 19, line 318: before the word: “which”.

,Please write shorter sentences. Some sentences are definitely too long and they can be divided into some shorter sentences, e.g., page 1, the sentence between the lines 16-19 or page 3, the sentence between the lines 88-90. Plese check it in the whole manuscript.

Page 2, line 74: please write “is also” instead of “also”.

Page 3, line 91: the commas before and after h(r,\thera, t) should be removed.

Page 3, line 98: the commas before and after \phi(r,\theta) should be removed.

Page 3, line 99: the word “or” would be better than “and” in the sentence: “Each function ...”.

Please remove comma before the word “and”:

page 3, line 101,

page 4, line 115,

page 5, line 132,

page 7, line 177,

page 7, line 179,

page 7, line 183.

Please check it in the whole manuscript.

Page 3, equation (10): It would be better to use the commas in the end of each line of this equation. The same thing in equations: (A-1), (Ą-3), (A-5), (A-7), (A-9).

Page 4, equation (11): It would be better to use the commas in the end of each line of this equation.

Page 4, equation (14): Please add comma in the end of the second line of this equation. The same thing in equations: (A-4), (A-8).

Page 4, equation (15) and page 5, equation (16): Please remove the dot at the present form and add the dot in the end of the second line of this formula.

Page 7,line 189: please remove comma before \Gamma.

Page 8, lines 192-193: please remove all brackets here, simply write the name of quantity and the denotation without brackets, e.g., the storage coefficient S.

Please write “boundary collocation points number” instead of “boundary collocation point number”:

page 8, line 203,

page 8, line 207,

page 8, line 208,

page 14, line 260,

page 16, line 287.

Please check it in the whole manuscript.

Please write “order of the basis functions” inteasd of “order of the basis function”:

page 8, line 205 (2 times),

page 8, line 208.

Please check it in the whole manuscript.

Page 8, figure 2: please check the titles of the x-axeses into: “order of the basis functions” and “number of boundary collocation points” for Fig. 2a and Fig. 2b, respectively.

In the sentence between the lines 224-226 it would be better for redears to write the values of the quantities next to the names of the quantities. Writting the values in the end of the sentence is inconvenient for readers if there is so many quantities.

Page 14, line 263, please write here rather “results” instead of “result”.

The authors write sometimes “Figure … displays”, e.g., page 14, line 264 or page 17, line 292. Please use rather the words “shows” or “illustrates” instead of “displays” in this context.

In equations (B-4), (B-10), (B-11) the commas or dots in the end of formulas should be more in the middle of the formula.

In conclusions, I recommend the manuscript to be published in the Water journal after minor revision.

Author Response

Comments from Reviewer 1#

The paper presents application of the Trefftz method for modeling subsurface flow with transient moving boundary. The authors presented very interesting and worthy of investigations subject in their manuscript. The results are very promising. I recommend the paper for minor revision. The subject of the article is within scope of the journal. The mathematical development is sound. I believe that the authors will find below some suggestions, which will help them to improve their manuscript:

Major comments:

Style of treating equations. Since the whole manuscript should be a text consisting of sentnces, the equation should not exist as something between two sentences, having nothing to do with these sentences – not beeing a part of these sentences. The equations should be the parts of the sentences. To do this, please:

h page 2, line 71: remove the dot,

Author’s reply:

Thank you so much for this invaluable comment. The manuscript has been revised at line 75 in page 2.

page 2, equation (1): add comma after R_t in the end of the formula,

Author’s reply:

Thank you so much for this invaluable comment. The comma has been added after R_t in the end of the formula in page 2.

page 2, line 72: please write the word “where” instead of “In the precending equations”, remove also the comma after this term – before R_t,

Author’s reply:

Thank you so much for this invaluable comment. The manuscript has been revised at line 76 in page 2.

page 2, line 78: remove the dot in the end of the sentence in this line,

Author’s reply:

Thank you so much for this invaluable comment. The manuscript has been revised at line 81 in page 2.

page 3, line 90: remove the dot in the end of the sentence in this line,

Author’s reply:

Thank you so much for this invaluable comment. The manuscript has been revised at line 93 in page 3.

page 3, equation (5): remove dot, write comma in this place,

Author’s reply:

Thank you so much for this invaluable comment. The manuscript has been revised at page 3.

page 3, line 91: please write the word “where” instead of “In the precending equations”, remove also the comma after this term – before \phi,

Author’s reply:

Thank you so much for this invaluable comment. The manuscript has been revised at line 94 in page 3.

page 3, line 93: remove the dot in the end of the sentence in this line,

page 3, line 97: remove the dot in the end of the sentence in this line,

Author’s reply:

Thank you so much for this invaluable comment. The manuscript has been revised at lines 95 and 99 in page 3.

page 3, equation (8): remove dot, write comma in this place,

Author’s reply:

Thank you so much for this invaluable comment. The manuscript has been revised in page 3.

page 3, line 98: please write the word “where” instead of “In the precending equations”, remove also the comma after this term – before R(r),

Author’s reply:

Thank you so much for this invaluable comment. The manuscript has been revised at line 3 in page 100.

page 4, line 114: remove the dot in the end of the sentence in this line,

page 4, line 117: remove the dot in the end of the sentence in this line,

page 4, line 126: remove the dot in the end of the sentence in this line,

page 5, line 129: remove the dot in the end of the sentence in this line,

Author’s reply:

Thank you so much for this invaluable comment. The manuscript has been revised at lines 116, 119, 128 and 131 in pages 4 and 5.

page 5, equation (18): add comma in the end of the formula,

Author’s reply:

Thank you so much for this invaluable comment. The comma has been added in the end of the formula (equation (18)) in page 5.

page 5, line 139: please remove the term: “, ad depicted in Equation (19),” (by the way, the term “depicted in Equation ...” is rather strange – equation is not a picture, please check it in the whole manuscript),

Author’s reply:

Thank you so much for this invaluable comment. The manuscript has been revised at line 141 in page 5.

page 5, line 139: remove the dot in the end of the sentence in this line,

page 7, line 176: remove the dot in the end of the sentence in this line,

Author’s reply:

Thank you so much for this invaluable comment. The manuscript has been revised at lines 141 and  178 in pages 5 and 7.

page 7, equation (27): please write comma instead of dot in the end of the formula in this equation,

Author’s reply:

Thank you so much for this invaluable comment. Equation (27) has been revised in page 7.

page 7, line 179: please write the word “where” instead of “In the precending equations”, remove also the comma after this term – before h^i(...),

Author’s reply:

Thank you so much for this invaluable comment. The manuscript has been revised at line 181 in page 7.

page 7, line 182: please remove the term “, as depicted in Equation (28).”, also the dot in the end of this term

Author’s reply:

Thank you so much for this invaluable comment. The manuscript has been revised at line 183 in page 7.

page 7, line 191: remove the dot in the end of the sentence in this line,

page 16, line 281: remove the dot in the end of the sentence in this line,

Author’s reply:

Thank you so much for this invaluable comment. The manuscript has been revised at lines 193 and 292 in pages 7 and 16.

page 21, line 377: please write “as follows” instead of “as following equations” and remove the dot.

Author’s reply:

Thank you so much for this invaluable comment. The manuscript has been revised at line 388 in page 21.

page 22, line 380: remove the dot in the end of the sentence in this line,

page 23, line 382: remove the dot in the end of the sentence in this line.

Please check now the whole manuscript in this context.

Author’s reply:

Thank you so much for this invaluable comment. The manuscript has been revised at lines 391 and 393 in pages 22 and 23.

Page 3, line 92: The authors wrote: ”Each function in equation (5) depends only on one of the variables r,\theta and t”. I'm not sure about that, because the function \phi depends on two variables: r and \theta. By the way, in the sentence the word “or” would be better than “and”.

Author’s reply:

Thank you so much for this invaluable comment. The manuscript has been revised at line 95 in page 3.

Page 3, line 103: why \lambda and \ksi are constant? Maybe it is your assumptions here. For example, it is easy to imagine that \Omega = 3*t, then \Omega' = 3 and consequently \lambda = 1/t.

Author’s reply:

Thank you so much for this invaluable comment. \lambda and \ksi are separation constants, which are adopted from the method of separation of variables.

The titles of the figures, tables, etc. should exist alone without the text of the manuscript. Thus please do not use abbreviations in these titles: page 8, figure 2, page 10, figure 4, page 13, figure 7 (by the way in these figure you does not present the MAE, because you present just absolute value, if I good understand MAE as the maximal value should be one value (the maximal ona) in the whole domain). Furthermore, maybe the absolute error should be also additionally defined.

Please check it in the whole manuscript.

Author’s reply:

Thank you so much for this invaluable comment. Figures 2, 4 and 7 have been revised in pages 8, 10 and 13.

Please give the mathematical formula for the error MAE. How it is defined? In how many test points the error was evaluated? Please write also in each example how many test points were used to evaluate the error.

Author’s reply:

Thank you so much for this invaluable comment. The mathematical formula for the error MAE has been given in Equation (33) in page 8. In addition, the test points of numerical examples have been mentioned at lines 220 and 246 in pages 8 and 10.

Minor comments:

Please use commas: page 1, lines 13-14: before the word “where”, page 1, line 15: before the word “in which” page 1, line 18: before the word “which”, page 1, line 21: before the word “because”, page 1, line 31: before the word “because”, page 2, line 57: before the word “where”, page 2, line 59: before the word: “in which”, page 2, line 62: before the word: “which”, page 19, line 310: before the word: “where”, page 19, line 317: before the word: “where”, page 19, line 318: before the word: “which”.

Author’s reply:

Thank you so much for this invaluable comment. The manuscript has been revised at lines 13-14, 15, 17, 21, 32, 60, 62, 65, 321, 328 and 329 in pages 1, 2 and 19.

Please write shorter sentences. Some sentences are definitely too long and they can be divided into some shorter sentences, e.g., page 1, the sentence between the lines 16-19 or page 3, the sentence between the lines 88-90. Please check it in the whole manuscript.

Author’s reply:

Thank you so much for this invaluable comment. The manuscript has been revised at lines 16-18 and 92-93 in pages 1 and 3.

Page 2, line 74: please write “is also” instead of “also”.

Author’s reply:

Thank you so much for this invaluable comment. The manuscript has been revised at line 78 in page 2.

Page 3, line 91: the commas before and after h(r,\thera, t) should be removed.

Author’s reply:

Thank you so much for this invaluable comment. The manuscript has been revised at line 94 in page 3.

Page 3, line 98: the commas before and after \phi(r,\theta) should be removed.

Author’s reply:

Thank you so much for this invaluable comment. The manuscript has been revised at line 100 in page 3.

Page 3, line 99: the word “or” would be better than “and” in the sentence: “Each function ...”.

Author’s reply:

Thank you so much for this invaluable comment. The manuscript has been revised at line 101 in page 3.

Please remove comma before the word “and”: page 3, line 101, page 4, line 115, page 5, line 132, page 7, line 177, page 7, line 179, page 7, line 183.

Please check it in the whole manuscript.

Author’s reply:

Thank you so much for this invaluable comment. The manuscript has been revised at lines 103, 117, 134, 179, 181 and 185 in pages 3, 4, 5 and 7.

Page 3, equation (10): It would be better to use the commas in the end of each line of this equation. The same thing in equations: (A-1), (Ą-3), (A-5), (A-7), (A-9).

Author’s reply:

Thank you so much for this invaluable comment. Equations (10), (A-1), (Ą-3), (A-5), (A-7) and (A-9) have been revised in pages 3, 19, 20 and 21.

Page 4, equation (11): It would be better to use the commas in the end of each line of this equation.

Author’s reply:

Thank you so much for this invaluable comment. Equations (11) has been revised in page 4.

Page 4, equation (14): Please add comma in the end of the second line of this equation. The same thing in equations: (A-4), (A-8).

Author’s reply:

Thank you so much for this invaluable comment. Equations (14), (A-4) and (A-8) have been revised in page 20.

Page 4, equation (15) and page 5, equation (16): Please remove the dot at the present form and add the dot in the end of the second line of this formula.

Author’s reply:

Thank you so much for this invaluable comment. Equations (15) and (16) have been revised in pages 4 and 5.

Page 7, line 189: please remove comma before \Gamma.

Author’s reply:

Thank you so much for this invaluable comment. The manuscript has been revised at line 191 in page 7.

Page 8, lines 192-193: please remove all brackets here, simply write the name of quantity and the denotation without brackets, e.g., the storage coefficient S.

Author’s reply:

Thank you so much for this invaluable comment. The manuscript has been revised at lines 194-195  in page 8.

Please write “boundary collocation points number” instead of “boundary collocation point number”: page 8, line 203, page 8, line 207, page 8, line 208, page 14, line 260, page 16, line 287.

Please check it in the whole manuscript.

Author’s reply:

Thank you so much for this invaluable comment. The manuscript has been revised at lines 203, 205, 212, 213, 271 and 298 in pages 8, 14 and 16.

Please write “order of the basis functions” instead of “order of the basis function”: page 8, line 205 (2 times), page 8, line 208.

Please check it in the whole manuscript.

Author’s reply:

Thank you so much for this invaluable comment. The manuscript has been revised at lines 209, 210 and 213 in page 8.

Page 8, figure 2: please check the titles of the x-axeses into: “order of the basis functions” and “number of boundary collocation points” for Fig. 2a and Fig. 2b, respectively.

Author’s reply:

Thank you so much for this invaluable comment. Figure 2a and Figure 2b have been revised in page 8.

In the sentence between the lines 224-226 it would be better for readers to write the values of the quantities next to the names of the quantities. Writting the values in the end of the sentence is inconvenient for readers if there is so many quantities.

Author’s reply:

Thank you so much for this invaluable comment. The manuscript has been revised at lines 233-235 in page 10.

Page 14, line 263, please write here rather “results” instead of “result”.

Author’s reply:

Thank you so much for this invaluable comment. The manuscript has been revised at line 273 in page 14.

The authors write sometimes “Figure … displays”, e.g., page 14, line 264 or page 17, line 292. Please use rather the words “shows” or “illustrates” instead of “displays” in this context.

Author’s reply:

Thank you so much for this invaluable comment. The manuscript has been revised at lines 274 and 303 in pages 14 and 17.

In equations (B-4), (B-10), (B-11) the commas or dots in the end of formulas should be more in the middle of the formula.

Author’s reply:

Thank you so much for this invaluable comment. Equations (B-4), (B-10), (B-11) have been revised in pages 22, 23 and 24.

In conclusions, I recommend the manuscript to be published in the Water journal after minor revision.

Author’s reply:

Thank you so much for this invaluable comment. The authors have carefully made their best efforts to revise the manuscript based on the comments. The authors sincerely hope that the revised manuscript could meet your requirement.

Reviewer 2 Report

Water review 648997

A spacetime meshless method for modeling subsurface flow with transient moving boundary

  Cheng-Yu Ku, Chih-Yu Liu, Jing-En Xiao, Weichung Yeih and Chia-Ming Fan

This paper presents a new  methodology  for solving transient nonlinear 2D moving boundary problems using Trefftz functions. Numerical results and comparisons from four  examples are provided to verify the merits of the proposed spacetime meshless method.

The paper is interesting and it is well written then I recommend to be considered for publication. I would like the authors to consider the following minor comments.

Include that Equation (1) is given in polar coordinates.

Define PDEs solved in examples 2, 3 and 4.

Please clarify the legend included in red color in Fig. 5 (a) .

Author Response

Comments from Reviewer 2#

This paper presents a new methodology for solving transient nonlinear 2D moving boundary problems using Trefftz functions. Numerical results and comparisons from four examples are provided to verify the merits of the proposed spacetime meshless method. The paper is interesting and it is well written then I recommend to be considered for publication. I would like the authors to consider the following minor comments.

Include that Equation (1) is given in polar coordinates.

Author’s reply:

Thank you so much for this invaluable comment. The manuscript has been revised at line 75 in page 2.

Define PDEs solved in examples 2, 3 and 4.

Author’s reply:

Thank you so much for this invaluable comment. The PDEs solved in examples 2, 3 and 4 have been mentioned in the revised manuscript. The manuscript has been revised at lines 189, 232, 261 and 291 in pages 7, 10, 14 and 16.

Please clarify the legend included in red color in Fig. 5 (a) .

Author’s reply:

Thank you so much for this invaluable comment. The legend included in red color in Figure 5 (a) denotes the origin. It is just for clearly identifying the location. Figure 5 (a) has also been revised in page 11.
